# Negative regulation of autophagy by UBA6-BIRC6–mediated ubiquitination of LC3

**Rui Jia, Juan S Bonifacino\***

Neurosciences and Cellular and Structural Biology Division, Eunice Kennedy Shriver National Institute of Child Health and Human Development National Institutes of Health, Bethesda, United States

**Abstract** Although the process of autophagy has been extensively studied, the mechanisms that regulate it remain insufficiently understood. To identify novel autophagy regulators, we performed a whole-genome CRISPR/Cas9 knockout screen in H4 human neuroglioma cells expressing endogenous LC3B tagged with a tandem of GFP and mCherry. Using this methodology, we identified the ubiquitin-activating enzyme UBA6 and the hybrid ubiquitin-conjugating enzyme/ubiquitin ligase BIRC6 as autophagy regulators. We found that these enzymes cooperate to monoubiquitinate LC3B, targeting it for proteasomal degradation. Knockout of UBA6 or BIRC6 increased autophagic flux under conditions of nutrient deprivation or protein synthesis inhibition. Moreover, UBA6 or BIRC6 depletion decreased the formation of aggresome-like induced structures in H4 cells, and α-synuclein aggregates in rat hippocampal neurons. These findings demonstrate that UBA6 and BIRC6 negatively regulate autophagy by limiting the availability of LC3B. Inhibition of UBA6/BIRC6 could be used to enhance autophagic clearance of protein aggregates in neurodegenerative disorders.

DOI: https://doi.org/10.7554/eLife.50034.001

**\*For correspondence:**
juan.bonifacino@nih.gov

**Competing interests:** The authors declare that no competing interests exist.

## Introduction

Macroautophagy (herein referred to as autophagy) is a catabolic process involving the engulfment of cytoplasmic materials into double-membraned autophagosomes that subsequently fuse with lysosomes to form autolysosomes, where the materials are degraded by acid hydrolases (*Bento et al., 2016*; *Dikic and Elazar, 2018*). Autophagy substrates include abnormal particles such as protein aggregates, damaged organelles and intracellular pathogens. Autophagy is also involved in the degradation of normal cellular constituents for survival under conditions of nutrient restriction or other stresses. Through these functions, autophagy plays crucial roles in the maintenance of cellular homeostasis. Defective autophagy contributes to the pathogenesis of various disorders, including neurodegeneration, cancer, cardiomyopathies and infectious diseases (*Dikic and Elazar, 2018*; *Levine and Kroemer, 2008*).

The mechanism of autophagy consists of multiple steps, including induction by cellular signals, phagophore formation and expansion, substrate engulfment, autophagosome closure, and autophagosome-lysosome fusion (*Braten et al., 2016*; *Dikic and Elazar, 2018*). Key components of the autophagy machinery are members of the Atg8 family of ubiquitin (Ub)-like proteins (LC3A, LC3B, LC3C, GABARAP, GABARAPL1 and GABARAPL2 in mammals), which play roles in autophagosome formation and autophagosome-lysosome fusion (*Wild et al., 2014*; *Johansen and Lamark, 2019*). The best studied member of this family is LC3B (product of the *MAP1LC3B* gene), which undergoes conversion from a soluble, cytosolic form (LC3B-I) to a phosphatidylethanolamine (PE)-conjugated, membrane-bound form (LC3B-II) (*Kabeya et al., 2004*). LC3B-II subsequently interacts with LC3-

interacting region (LIR) motifs of various cargo receptors to capture autophagic cargos into forming autophagosomes (*Birgisdottir et al., 2013*). Among these receptors are cytosolic proteins such as SQSTM1 (p62), NBR1, NDP52, OPTN and TAX1BP1, which bind polyubiquitinated cargos via their Ub-binding domains (*Birgisdottir et al., 2013*; *Wild et al., 2014*; *Johansen and Lamark, 2019*). Other cargo receptors are anchored to the autophagic cargos via their transmembrane domains, as is the case for BNIP3, NIX and FUNDC1 in mitochondrial autophagy (mitophagy) (*Birgisdottir et al., 2013*; *Wild et al., 2014*; *Johansen and Lamark, 2019*), and RTN3, SEC62, CCPG1, FAM134B and TEX264 in endoplasmic reticulum (ER) autophagy (ER-phagy) (*Khaminets et al., 2015*; *Fumagalli et al., 2016*; *Grumati et al., 2017*; *Smith et al., 2018*; *Chino et al., 2019*; *Johansen and Lamark, 2019*). After fusion of autophagosomes with lysosomes, the autophagy cargos, together with the Atg8-family proteins and cargo receptors, are degraded in lysosomes (*Tanida et al., 2005*; *Bjørkøy et al., 2005*). In addition to participating in cargo recruitment to the forming autophagosome, LC3B interacts with the LIR motif of FYCO1 (FYVE and coiled-coil domain containing protein 1), which serves as an adaptor to the kinesin-1 motor, enabling anterograde transport of autophagosomes along microtubule tracks (*Pankiv et al., 2010*). Furthermore, LC3B interacts with the LIR motif of another adaptor protein, PLEKHM1 (pleckstrin homology domain containing protein family member 1), which functions as a tether in autophagosome-lysosome fusion (*McEwan et al., 2015*).

The autophagy machinery is regulated by post-translational modifications such as phosphorylation and ubiquitination. Several kinases have been implicated in positive or negative regulation of autophagy. As an example of positive regulation, the unc-51-like autophagy-activating kinase 1 (ULK1) phosphorylates the VPS34 (*Egan et al., 2015*; *Russell et al., 2013*), BECN1 (*Russell et al., 2013*) and ATG14L1 (*Park et al., 2016*) components of the class III PI3K complex, which subsequently catalyzes the conversion of phosphatidylinositol (PI) to phosphatidylinositol 3-phosphate [PI(3)P], thus triggering phagophore formation. ULK1 itself is activated by phosphorylation on Ser-317, Ser-555 and Ser-777 by AMP-activated protein kinase (AMPK) (*Egan et al., 2011*; *Kim et al., 2011*). On the other hand, the mechanistic target of rapamycin (mTOR) complex-1 (mTORC1) kinase negatively regulates autophagy by phosphorylating ULK1 on Ser-757, and thus preventing the interaction of ULK1 with AMPK (*Kim et al., 2011*). The mTORC1 kinase exerts an additional inhibitory effect on autophagy by phosphorylating the autophagy protein UVRAG, a modification that decreases autophagosome maturation and autophagosome-lysosome fusion (*Kim et al., 2015*).

Ubiquitination also plays positive and negative roles in autophagy. In fact, some of the kinases or kinase complexes that regulate autophagy are themselves targets of ubiquitination. For instance, ULK1 and BECN1 are positively regulated by polyubiquitination mediated by the Ub-ligase (E3) TRAF6 (*Shi and Kehrl, 2010*; *Nazio et al., 2013*). Non-enzymatic components of the autophagy machinery can also be positively regulated by ubiquitination, as is the case for the polyubiquitination of OPTN by the E3 HACE1, which promotes assembly of OPTN with SQSTM1 and thus results in enhanced autophagic degradation (*Liu et al., 2014*). An example of the negative effects of ubiquitination on autophagy is the activation of mTORC1 by TRAF6-mediated polyubiquitination of the catalytic mTOR subunit, with consequent inhibition of autophagy (*Linares et al., 2013*). Furthermore, ULK1 and components of the class III PI3K complex are targeted for polyubiquitination and degradation by the E3 Cul3-KLHL20 (*Liu et al., 2016*). Finally, BECN1 is polyubiquitinated by the RNF216 and NEDD4 E3s, causing destabilization of the class III PI3K complex (*Platta et al., 2012*; *Xu et al., 2014*).

Both phosphorylation and ubiquitination have been targeted for pharmacologic manipulation of autophagy. For example, inhibition of mTORC1 by the drugs rapamycin or torin1 (*Noda and Ohsumi, 1998*; *Thoreen et al., 2009*), or activation of AMPK by metformin (*Shi et al., 2012*), are widely used to stimulate autophagy. In contrast, inhibition of the class III PI3K complex by wortmannin, LY294002 or 3-methyladenine impairs autophagy (*Blommaart et al., 1997*; *Wu et al., 2010*). Finally, inhibition of USP10/13-mediated deubiquitination of BECN1 by spautin-1 causes destabilization and degradation of the class III PI3K complex, decreasing autophagy (*Liu et al., 2011*). Autophagy stimulation has been proposed as a possible therapy for neurodegenerative disorders caused by accumulation of intracellular protein aggregates (*Liu et al., 2011*). Autophagy inhibition, on the other hand, could be used to treat hyperproliferative disorders fueled by increased autophagy, as is the case for some cancers (*Jiang et al., 2015*). However, the involvement of known regulators of autophagy in other cellular processes has hampered efforts to exploit their pharmacologic activation

or inhibition for clinical uses. Therefore, there is a need to identify additional autophagy regulators that could be targeted for therapeutic applications.

To identify additional autophagy regulators, we conducted a genome-wide CRISPR/Cas9 knock-out (KO) screen using cells that were gene-edited to express endogenous LC3B tagged with tandem GFP and mCherry fluorescent proteins. This screen is based on the acid-induced quenching of GFP, but not mCherry, upon fusion of autophagosomes with lysosomes (*Kimura et al., 2007*). Mutant cells displaying higher GFP:mCherry fluorescence ratios were selected by fluorescence-activated cell sorting (FACS), and the mutated genes were identified by next-generation sequencing. The reliability of the screen was demonstrated by the identification of genes encoding virtually all core components of the autophagy machinery. In addition, we identified other candidates, among which were two ubiquitination-related proteins: the Ub-activating enzyme (E1) UBA6 and the hybrid Ub-conjugating enzyme/Ub-ligase (E2/E3) BIRC6. Further analyses showed that these enzymes cooperate to catalyze monoubiquitination of LC3B, marking it for degradation by the proteasome. KO of UBA6 or BIRC6 increased the levels of LC3B, and lessened the accumulation of aggresome-like induced structures (ALIS) in non-neuronal cells and α-synuclein aggregates in neurons. These findings thus identify UBA6 and BIRC6 as negative regulators of autophagy, whose inhibition could be used to enhance autophagy and thus prevent the accumulation of pathogenic protein aggregates.

## Results

### Identification of autophagy genes by CRISPR/Cas9 KO screening in human cells

This project was aimed at identifying novel autophagy genes in mammalian cells by taking advantage of the development of pooled CRISPR/Cas9 KO libraries for whole-genome phenotypic screens (*Joung et al., 2017*). To this end, we developed an indicator cell line that enabled the selection of autophagy-defective cells by FACS. This cell line, named H4-tfLC3B (H4-tandem fluorescent LC3B), was made by gene-editing human neuroglioma H4 cells to express endogenous LC3B tagged at its N-terminus with tandem GFP and mCherry fluorescent proteins (designated GFP-mCherry-LC3B) (*Figure 1—figure supplement 1A and B*). The quenching of GFP, and not mCherry, in the acidic pH of autolysosomes allows the use of the GFP:mCherry ratio to monitor the delivery of GFP-mCherry-LC3B from neutral autophagosomes to acidic autolysosomes (process referred to as 'autophagy flux') (*Kimura et al., 2007*). Moreover, endogenous tagging avoids the overexpression artifacts caused by transfection of cells with plasmids encoding fluorescently tagged LC3B (*Kimura et al., 2007*). Live-cell imaging of H4-tfLC3B cells revealed puncta displaying both GFP and mCherry fluorescence (*i.e.*, autophagosomes), or only mCherry fluorescence (*i.e.*, autolysosomes) (*Figure 1A*). Treatment of these cells with the vacuolar ATPase (v-ATPase) inhibitor bafilomycin A$_1$ increased the number of puncta containing GFP in addition to mCherry (*Figure 1A*), as well as the GFP:mCherry ratio in FACS analyses (seen as a displacement to the right in *Figure 1B*), consistent with the presence of quenched GFP in autolysosomes. Likewise, siRNA-mediated knock down (KD) of the autophagy protein ATG7 increased the GFP:mCherry ratio (*Figure 1—figure supplement 1C and D*), in accordance with the requirement of ATG7 for conversion of LC3B-I to LC3B-II and subsequent delivery of LC3B-II to autolysosomes. In contrast, depletion of serum and amino acids from the medium (*i.e.*, nutrient deprivation) decreased the GFP:mCherry ratio detected by FACS (seen as a shift to the left in *Figure 1C*, *Figure 1—figure supplement 1E*), as expected from the induction of autophagy under conditions of nutrient shortage (*Kuma et al., 2004*). Thus, the H4-tfLC3B cell line exhibited the expected responses to the inhibition or stimulation of autophagy.

H4-tfLC3 cells were mutated with the human GeCKO v2 lentiviral pooled library, containing 123,411 sgRNAs targeting 19,050 genes and 1864 miRNAs (*Sanjana et al., 2014*) (*Figure 1D*). FACS was used to collect the top 1% of cells with increased GFP:mCherry ratio. These cells were successively expanded and sorted an additional three rounds, resulting in an enrichment of cells with high GFP:mCherry ratio from 1.02% to 89.0% (*Figure 1E*, *Figure 1—figure supplement 1F*). The genes that were mutated in these cells were identified by next-generation sequencing of the integrated sgRNAs (*Figure 1D*, *Supplementary file 1*). The enrichment of each gene in the sorted *vs.* unsorted population was calculated using the MAGeCK algorithm (*Li et al., 2014*) (*Figure 1F*). Most of the top-ranked genes in this analysis corresponded to known components of the core autophagy

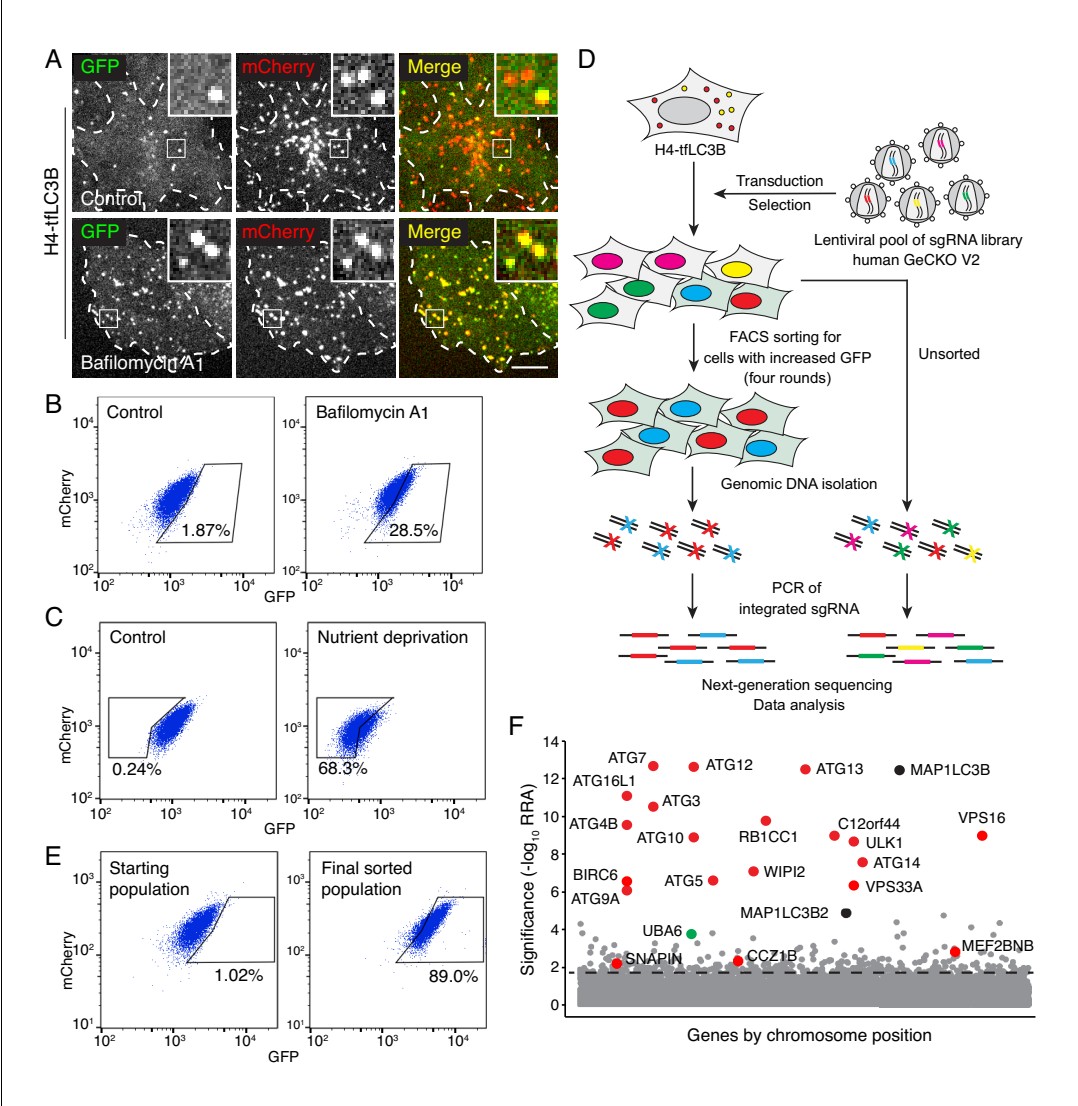

**Figure 1.** Genome-wide CRISPR/Cas9 screen for the identification of autophagy genes. (A) Live-cell imaging of H4 cells expressing GPF-mCherry-LC3B (H4-tfLC3B). Cells were incubated in the absence (control) or presence of 50 nM bafilomycin A₁ for 2 hr prior to imaging. Single-channel images are shown in grayscale. Cell edges are outlined. Scale bar: 10 μm. Insets are 3.6x magnifications of the boxed areas. (B) H4-tfLC3B cells were incubated without (control) or with 50 nM bafilomycin A₁ for 2 hr, and GFP and mCherry fluorescence was measured by FACS. (C) H4-tfLC3B cells were incubated in regular medium (control) or amino-acid- and serum-free medium for 4 hr (nutrient deprivation), and then analyzed by FACS. (D) Schematic representation of the genome-wide CRISPR/Cas9 screen. H4-tfLC3B cells were mutated with a pooled lentiviral GeCKO v2 library. Cells with high GFP: mCherry ratio were sorted and propagated; after four rounds of sorting, genomic DNA was isolated. The sequences of sgRNAs were determined by next-generation sequencing. (E) FACS analysis of cells infected with the lentiviral pool (starting population) and cells after four rounds of sorting (final sorted population). In B, C and E, the percentages of cells in the boxed areas are indicated. (F) Ranking of genes from the CRISPR/Cas9 screen based on the RRA (Robust Ranking Aggregation) algorithm score calculated using the MAGeCK method. Genes known to participate in autophagy are labeled in red; MAP1LC3B genes (i.e., LC3B and LC3B2) are labeled in black; a gene not previously implicated in autophagy, UBA6, is labeled in green. The identification of LC3B and LC3B2 may be due to the synthesis of a truncated GPF-mCherry-LC3B that cannot be degraded. The genes above the horizontal dotted line were tested in the secondary screen.

DOI: https://doi.org/10.7554/eLife.50034.002

The following figure supplement is available for figure 1:

**Figure supplement 1.** Generation, analysis and mutant selection of H4 cells expressing endogenously-tagged GFP-mCherry-LC3B.

DOI: https://doi.org/10.7554/eLife.50034.003

machinery (*Figure 1F*, *Supplementary file 2*) (*Bento et al., 2016*), demonstrating the reliability of the screen. A number of previously reported autophagy effectors and regulators were also identified with lower but still significant scores; these included the MEF2BNB and SNAPIN subunits of BORC (*Pu et al., 2015*; *Jia et al., 2017*), C18orf8 (*Vaites et al., 2017*), BECN1 (*Liang et al., 1999*), RAB1A (*Webster et al., 2016*), EPG5 (*Wang et al., 2016*) and CCZ1B (*Vaites et al., 2017*) (*Figure 1F*, *Supplementary file 2*).

To confirm candidates from the primary screen, we constructed a secondary pooled CRISPR/Cas9 lentiviral library targeting the top 432 protein-coding genes, by combining the sgRNA sequences from two published CRISPR/Cas9 screens (*Joung et al., 2017*; *Wang et al., 2015*) (*Supplementary file 3*). This secondary library excluded sgRNAs for the top 20 autophagy genes, except the sgRNAs for ATG7 and VPS16, which were included as positive controls (*Supplementary file 3*). H4-tfLC3B cells were mutated by the secondary CRISPR/Cas9 library, and the cells with defective autophagy were selected by three rounds of FACS (*Figure 2A*, *Figure 2— figure supplement 1A*), and analyzed by next-generation sequencing (*Figure 2B*, *Supplementary files 4*, *5*), as described above. Most of the genes exhibiting significant scores in this secondary screen were also known regulators of autophagy (*Figure 2B*). *UBA6*, however, stood out as a gene with relatively high scores in both the primary (*Figure 1F*) and secondary screens (*Figure 2B*), and that had not been previously implicated in autophagy. This finding prompted us to investigate the role of UBA6 in autophagy.

## UBA6 depletion increases the levels of LC3B-I without reducing the levels of LC3B-II

UBA6 (ubiquitin-like modifier activating enzyme 6) encodes an E1 Ub-activating enzyme having a cysteine residue that forms a thioester bond with the C-terminal glycine of Ub (*Groettrup et al., 2008*). To investigate the involvement of UBA6 in autophagy, we initially used siRNAs to KD UBA6 in H4 and HeLa cervical carcinoma cells, and examined the levels of the cytosolic (LC3B-I) and membrane-bound (LC3B-II) forms of LC3B as indicators of autophagy status (*Kabeya et al., 2004*) (*Figure 2C and D*). Interestingly, we observed that UBA6 KD caused 3–4–fold increases in the levels of LC3B-I without altering the levels of LC3B-II in both cell lines (*Figure 2C and D*). This was in contrast to ATG7 KD, which increased the levels LC3B-I but decreased the levels of LC3B-II (*Figure 2— figure supplement 1B and C*). CRISPR/Cas9 KO of UBA6 also resulted in increased levels of LC3B-I and unchanged levels of LC3B-II, a phenotype that was partially reversed by transfection with plasmids encoding UBA6 tagged at either the N- or C-terminus with the MYC epitope (*Figure 2E and F*). UBA6 KO caused a similar increase in the levels of LC3A-I but not of other Atg8-family members such as GABARAP and GABARAPL1 (*Figure 2—figure supplement 1D*). UBA6 KO did not cause noticeable changes in the levels of other components of the autophagy machinery, including components of the LC3-conjugation system (ATG3, ATG5, ATG7, ATG12, ATG16L1), WIPI2, the PI3K complex (BECN1, p-BECN1 and ATG14), the ULK1 complex (ATG13, p-ATG13, ULK1, p-ULK1), and the mTOR signaling pathway (S6K, p-S6K, 4EBP, p-4EBP, TSC2, p-TSC2, AKT, p-AKT) (*Bento et al., 2016*) (*Figure 2—figure supplement 1E*). Likewise, fluorescent staining for the autophagy machinery components SQSTM1, WIPI2 and ATG9A (*Figure 2—figure supplement 2A–C*), and for markers of the ER (calnexin), Golgi complex (GM130), mitochondria (TOM20), early endosomes (RAB5, APPL1, EEA1), lysosomes (LAMP1, LAMTOR4) and the cytoskeleton (β-tubulin, F-actin) (*Figure 2— figure supplement 2D–J*), showed no differences in UBA6-KO cells relative to the parental H4 cells. In addition, we observed that UBA6-KO cells proliferated at the same rate as WT cells, with doubling times of ~27 hr (*Figure 2—figure supplement 3A*). Bafilomycin A$_1$ treatment caused a similar accumulation of LC3B-II in WT, UBA6-KO cells and UBA6-KO cells rescued by transfection with a plasmid encoding UBA6-MYC (*Figure 2G and H*, *Figure 2—figure supplement 3B and C*), indicating that autophagy initiation and LC3B conjugation operated normally in UBA6-KO cells. Finally, WT and UBA6-KO cells expressing GFP-mCherry-LC3B and loaded with internalized Alexa Fluor 647-conjugated dextran exhibited similar ratios of autophagosomes (*i.e.*, puncta positive for GFP and mCherry, but negative for Alexa Fluor 647) to autolysosomes (*i.e.*, puncta positive for mCherry and Alexa Fluor 647, but negative for GFP) (*Figure 2I and J*), indicating that UBA6 KO had no effect on autophagosome-lysosome fusion and lysosome acidification. From all of these experiments we concluded that UBA6 depletion increased LC3B-I levels without affecting its conversion to LC3B-II or its eventual degradation in lysosomes.

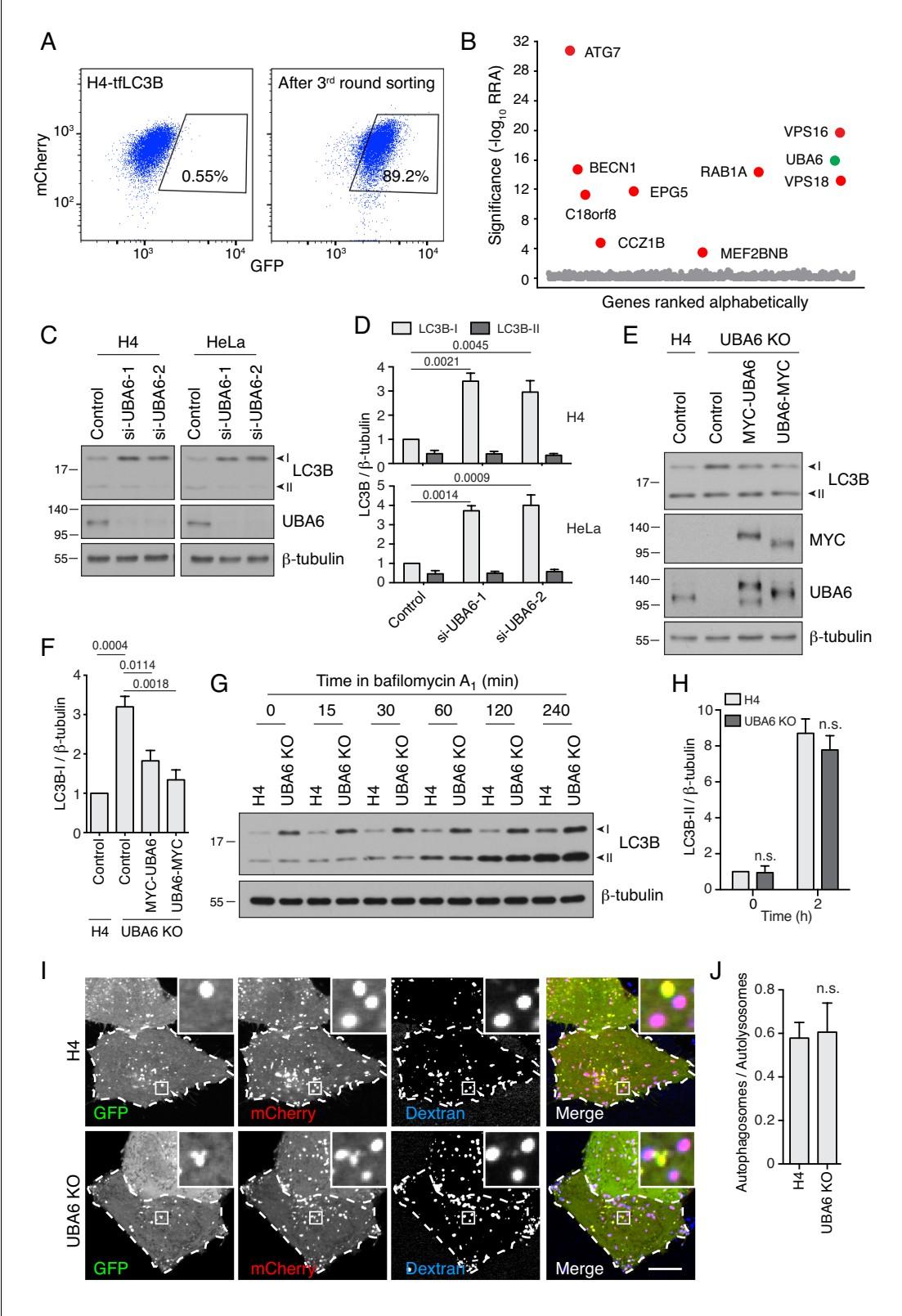

**Figure 2.** UBA6-KO cells accumulate LC3B-I but not LC3B-II. (**A**) FACS analysis of cells from the secondary screen showing enrichment of library-infected H4 cells with increased GFP fluorescence after three rounds of sorting. The percentages of cells in the boxed areas are indicated. (**B**) Ranking of genes in the CRISPR/Cas9 screen based on RRA. Genes highlighted in red were previously reported to function in autophagy. The novel autophagy regulator UBA6 is highlighted in green. (**C**) H4 or HeLa cells were transfected with control or either of two UBA6 siRNAs. After 48 hr, cells were analyzed

*Figure 2 continued on next page*

*Figure 2 continued*

by SDS-PAGE and immunoblotting for LC3B, UBA6 and β-tubulin (control). In this and all other relevant figures, the positions of the I and II forms of LC3B are indicated. (**D**) Quantification of the ratio of LC3B-I and -II to β-tubulin. The ratio for control siRNA was arbitrarily set at 1. Values are the mean ± SEM from three independent experiments such as that shown in C. The indicated *p*-values relative to the control were calculated using a two-way ANOVA with Tukey's multiple comparisons test. (**E**) WT or UBA6-KO H4 cells were transfected with control plasmid or plasmids encoding MYC-UBA6 or UBA6-MYC. The cells were analyzed by SDS-PAGE and immunoblotting with antibodies to the antigens on the right. (**F**) Quantification of ratio of LC3B-I to β-tubulin. The ratio for the control plasmid transfection in WT H4 cells was arbitrarily set at 1. Values are the mean ± SEM from three independent experiments. The indicated *p*-values were calculated using a one-way ANOVA with Dunnett's multiple comparisons test. (**G**) WT and UBA6-KO H4 cells were incubated with 50 nM bafilomycin A$_1$ for the indicated periods prior to SDS-PAGE and immunoblotting with antibodies to LC3B and β-tubulin. In C, E and G, the positions of molecular mass markers (in kDa) are indicated on the left. (**H**) Quantification of the ratio of LC3B-II to β-tubulin. The ratio of WT H4 cells not treated with bafilomycin A$_1$ was arbitrarily set at 1. Values are the mean ± SEM from three independent experiments such as that shown in G. *p*-values were calculated using two-way ANOVA with Tukey's multiple comparisons tests. n.s.: not significant. (**I**) WT and UBA6-KO H4 cells were transfected with a plasmid encoding GFP-mCherry-LC3B and allowed to internalize Alexa Fluor 647-conjugated dextran for 16 hr at 37°C to label late endosomes, lysosomes and autolysosomes. GFP (green), mCherry (red) and Alexa Fluor 647 (blue) fluorescence was visualized by live-cell imaging. Single-channel images are shown in grayscale. Cell edges are outlined. Scale bar: 10 μm. Insets are 4.6x magnifications of the boxed areas. (**J**) The ratio of autophagosomes (red-green–positive puncta) to autolysosomes (red-blue–positive puncta) was determined. Bars represent the mean ± SEM of the ratio in 20 cells from three independent experiments. N.s.: not significant, according to an unpaired Student's *t* test.

DOI: https://doi.org/10.7554/eLife.50034.004

The following figure supplements are available for figure 2:

**Figure supplement 1.** Secondary screen for autophagy mutants and expression of autophagy proteins in WT and UBA6-KO H4 cells.
DOI: https://doi.org/10.7554/eLife.50034.005

**Figure supplement 2.** Distribution of organellar proteins in WT and UBA6-KO H4 cells.
DOI: https://doi.org/10.7554/eLife.50034.006

**Figure supplement 3.** Proliferation rate and LC3B mRNA expression in UBA6-KO and BIRC6-KO cells.
DOI: https://doi.org/10.7554/eLife.50034.007

## UBA6 mediates ubiquitination and proteasomal degradation of LC3B

We next examined the mechanism by which UBA6 depletion increased the levels of LC3B-I. Real-time quantitative PCR (RT-qPCR) showed that KO of UBA6 did not change the levels of LC3B mRNA (*Figure 2—figure supplement 3D*). Since UBA6 is an E1 enzyme (*Groettrup et al., 2008*), we then hypothesized that it could decrease the levels of LC3-I protein by mediating LC3B-I ubiquitination and proteasomal degradation. Indeed, treatment with the proteasome inhibitor MG132 resulted in a ~3 fold increase in the level of LC3B-I in WT H4 cells, but did not change the already elevated level of LC3B-I in UBA6-KO cells (*Figure 3A and B*). In addition, an in vivo ubiquitination assay showed that transgenic FLAG-tagged LC3B became conjugated with a single copy of HA-tagged Ub (*Figure 3C*). The level of monoubiquitinated FLAG-LC3B was increased by treatment with MG132 (*Figure 3C*) and decreased by KO of UBA6 (*Figure 3D*). The fraction of FLAG-LC3B that was monoubiquitinated was low, as observed on a long exposure of the anti-FLAG blot for MG132-treated cells (*Figure 3C*, arrow), although this could be due to the action of deubiquitinating enzymes (*Clague et al., 2019*). Mutation of LC3B-I glycine-120 – the LC3B-I residue that is conjugated to phosphatidylethanolamine to produce LC3B-II (*Kabeya et al., 2004*) – to alanine prevented the association of LC3B with autophagosomes (*Figure 3E*) without inhibiting ubiquitination (*Figure 3F*), indicating that LC3B can be ubiquitinated in the cytosol. The in vivo ubiquitination assay was additionally used to show that FLAG-tagged LC3A and LC3C were also monoubiquitinated, but GABARAP, GABARAPL1 and GABARAPL2 were not (*Figure 3G*). Finally, we found that endogenous LC3B was also modified with a single HA-Ub moiety (*Figure 3H*). Taken together, these experiments indicated that UBA6 promotes monoubiquitination of cytosolic LC3B-I, targeting it for degradation by the proteasome.

To identify the ubiquitination site on LC3B, we mutated each of its 10 lysines (*Figure 4A*) to arginine, a residue that cannot be conjugated to Ub. We observed that double K49R-K51R and single K51R mutants were not ubiquitinated, indicating that lysine-51 was the main ubiquitination site on LC3B (*Figure 4B and C*). Inspection of the structure of LC3B showed that lysine-51 is part of a hydrophobic pocket involved in the recognition of a LIR motif from the autophagy receptor SQSTM1 (*Figure 4D*) (*Ichimura et al., 2008*). Indeed, co-immunoprecipitation analyses showed that the K51R substitution abolished the interaction of FLAG-LC3B with endogenous forms of SQSTM1 and

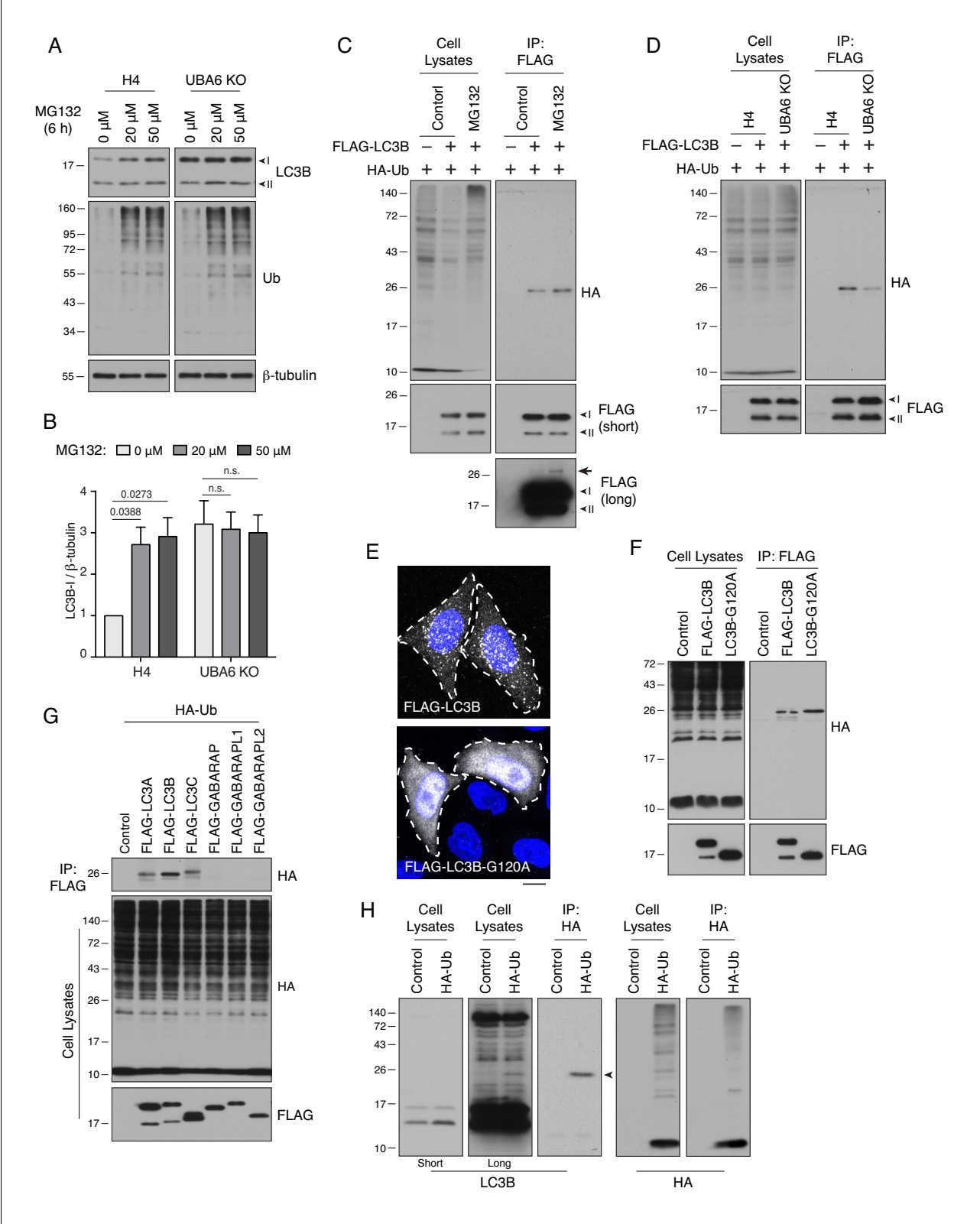

**Figure 3.** UBA6 participates in monoubiquitination of LC3B. (**A**) WT and UBA6-KO H4 cells were incubated with 0, 20 or 50 µM MG132 for 6 hr, and analyzed by SDS-PAGE and immunoblotting with antibodies to LC3B, Ub and β-tubulin (loading control). (**B**) The ratio of LC3B-I to β-tubulin was determined from experiments such as that in A. The ratio in WT H4 cells without MG132 treatment was arbitrarily set at 1. Bars represent the mean ± SEM of the ratio from three independent experiments. The indicated *p*-values were calculated using two-way ANOVA with Tukey's multiple

*Figure 3 continued on next page*

*Figure 3 continued*

comparisons tests. (C) WT and UBA6-KO cells were transfected with plasmids encoding HA-Ub and FLAG-LC3B. Cell lysates were immunoprecipitated (IP) with antibody to the FLAG epitope, and cell lysates and immunoprecipitates were analyzed by SDS-PAGE and immunoblotting with antibodies to the HA and FLAG epitopes. Ubiquitinated FLAG-LC3B can be seen as a faint band on the long exposure of the anti-FLAG blot (arrow). (D) H4 cells were transfected with plasmids encoding HA-Ub and FLAG-LC3B. Cells were incubated with 20 µM MG132 for 6 hr before lysis and immunoprecipitation with antibody to the FLAG epitope. Cell lysates and immunoprecipitates were analyzed by SDS-PAGE and immunoblotting with antibodies to the HA and FLAG epitopes. (E) Immunofluorescence microscopy showing the localization of FLAG-LC3B and FLAG-LC3B-G120A mutant in WT H4 cells. DAPI (blue) was used to stain the nucleus. Cell edges are outlined. Scale bar: 10 µm. (F) H4 cells expressing HA-Ub together with FLAG-LC3B or FLAG-LC3B-G120A mutant were analyzed by immunoprecipitation with antibody to the FLAG epitope, followed by SDS-PAGE and immunoblotting with antibodies to the HA and FLAG epitopes. (G) H4 cells were transfected with plasmids encoding FLAG-tagged Atg8-family proteins and HA-Ub. Cell lysates were subjected to immunoprecipitation with antibody to the FLAG epitope, and cell lysates and immunoprecipitates were analyzed by SDS-PAGE and immunoblotting with antibodies to the HA and FLAG epitopes. (H) H4 cells were transfected with control or HA-Ub-encoding plasmids. Ubiquitinated proteins in the cell lysates were enriched by immunoprecipitation with antibody to the HA epitope. Cell lysates and immunoprecipitates were analyzed by SDS-PAGE and immunoblotting with antibodies to LC3B or the HA epitope. In A, C, D, G, F and H, the positions of molecular mass markers (in kDa) are indicated on the left.

DOI: https://doi.org/10.7554/eLife.50034.008

another autophagy receptor, NBR1 (*Figure 4E*). These results thus confirmed the critical function of lysine-51 in the recognition of autophagy receptors, and suggested that an additional function of LC3B ubiquitination by UBA6 may be to inhibit this recognition.

## BIRC6 acts like UBA6 to decrease the levels of LC3B-I

E1 enzymes function together with E2 Ub-conjugating enzymes and E3 Ub-ligases to add Ub to substrates (*Pickart, 2001*). To identify the E2 and E3 responsible for LC3B ubiquitination, we conducted another CRISPR/Cas9 KO screen using a lentiviral library of 10,108 sgRNAs targeting all major E1, E2, E3 and deubiquitinating enzymes, as well as other ubiquitination-related proteins (*Supplementary file 6*). As done before, H4-tfLC3B cells were infected with this library, and cells with high GFP:mCherry ratios were enriched by three rounds of FACS (*Figure 5A*, *Figure 5—figure supplement 1A*). Next-generation sequencing and MAGeCK analysis revealed that KO of BIRC6 [baculoviral IAP (inhibitor of apoptosis) repeat containing 6] inhibited quenching of GFP even more potently than UBA6 KO (*Figure 5B*, *Supplementary files 7*, *8*), as already seen in the primary screen (*Figure 1F*). The BIRC6 product (also known as BRUCE and APOLLON) is a giant, 528 kDa protein containing an N-terminal BIR (Baculoviral IAP Repeat) domain and a C-terminal UBC (Ub Conjugation) domain (*Bartke et al., 2004*) (*Figure 6—figure supplement 1A*). Previous work showed that BIRC6 combines, in a single polypeptide, E2 and E3 activities towards substrates such as SMAC (second mitochondria-derived activator of caspases) and caspase 9, which are involved in the regulation of apoptosis (*Bartke et al., 2004*; *Hao et al., 2004*). In addition, BIRC6 was shown to promote the progression of prostate, liver and colorectal cancers (*Low et al., 2013*; *Ren et al., 2005*; *Hu et al., 2015*; *Tang et al., 2015*).

In further experiments, we found that depletion of BIRC6 by KD or KO in H4 cells increased LC3B-I levels even more dramatically than depletion of UBA6, in both cases with no change in the levels of LC3B-II (*Figure 5C–5E*). Fractions of the total LC3B were previously shown to exist in association with cytoplasmic protein aggregates (*Kuma et al., 2007*), nuclei (*Huang et al., 2015*) and microtubules (*Mann and Hammarback, 1994*). We observed that, in H4 cells grown under normal culture conditions, the majority of LC3B-I and -II was extractable with Triton-X-100 and therefore not associated with protein aggregates (*Figure 5—figure supplement 1B*). We did, however, find that about half of LC3B-I and -II was present in a nuclear fraction (*Figure 5—figure supplement 1C*), as previously reported (*Huang et al., 2015*). We did not detect any association of LC3B-I and -II with microtubules (*Figure 5—figure supplement 1D*). Regardless of the fractions in which LC3B was present, KO of BIRC6 resulted in increased levels of LC3-I (*Figure 5—figure supplement 1B–1D*). BIRC6-KO cells also exhibited similar growth rates (*Figure 2—figure supplement 3A*) and LC3B mRNA levels (*Figure 2—figure supplement 3D*) relative to WT cells. Incubation with bafilomycin A$_1$ caused a similar accumulation of LC3B-II in WT, UBA6-KO and BIRC6-KO cells (*Figure 5—figure supplement 1E and F*). Likewise, the ratio of autophagosomes to autolysosomes determined by fluorescence microscopy, as described above, was unchanged in BIRC6-KO relative to WT cells

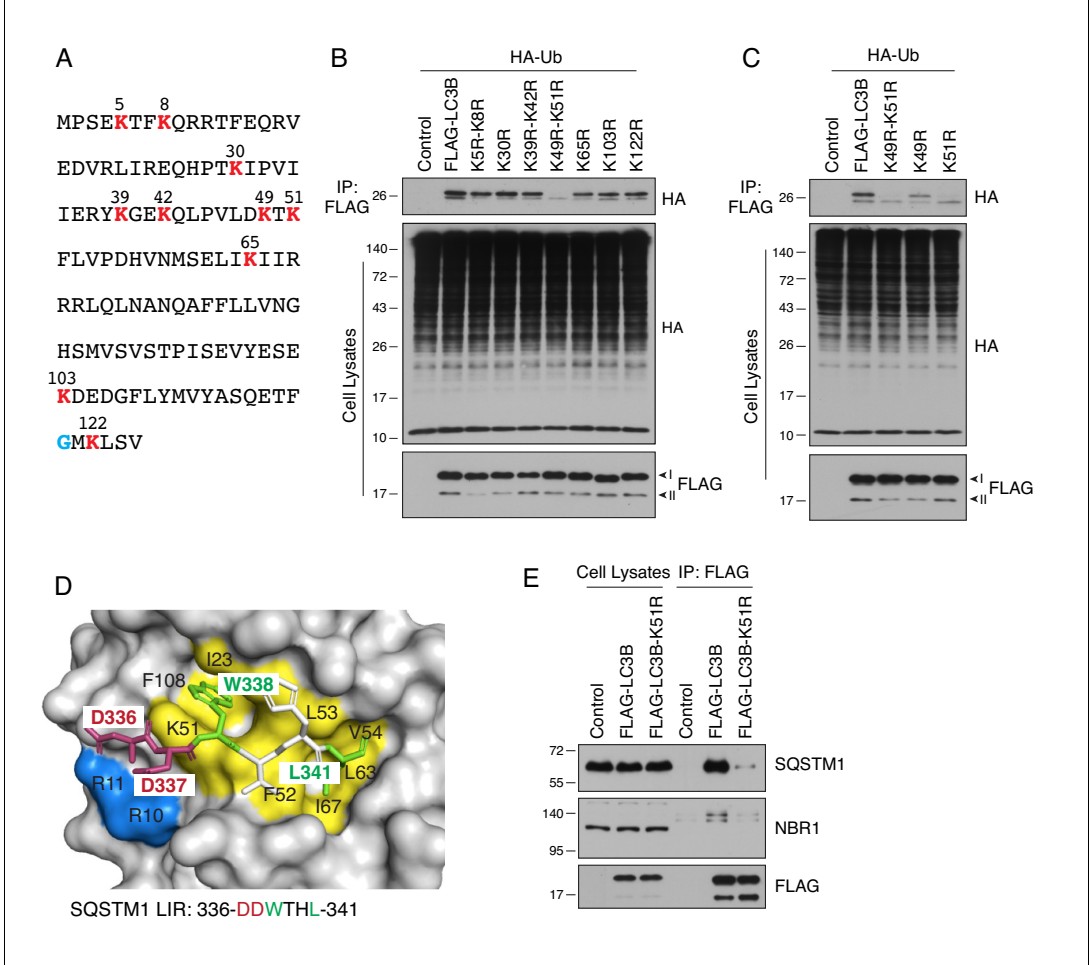

**Figure 4.** LC3B is ubiquitinated on lysine-51. (**A**) Sequence of human LC3B. Lysine residues are highlighted in red. A glycine residue that becomes conjugated to phosphatidylethanolamine is highlighted in blue. (**B,C**) H4 cells were transfected with plasmids encoding WT and mutant LC3B constructs and HA-Ub. The ubiquitination of LC3B was examined as described for **Figure 3C**. (**D**) Binding of the LIR motif of SQSTM1 (DDWTHL) (stick representation) to LC3B (surface representation) (PDB code: 2ZJD). Yellow and green indicate residues involved in electrostatic interactions; blue and red indicate residues forming hydrogen bonds. (**E**) H4 cells were transfected with plasmids encoding control, FLAG-LC3B or FLAG-LC3-K51R mutant plasmids. Cell lysates were immunoprecipitated with antibody to the FLAG epitope. Cell lysates and immunoprecipitates were analyzed by SDS-PAGE and immunoblotting with antibodies to SQSTM1, NBR1 and the FLAG epitope. In B, C and E, the positions of molecular mass markers (in kDa) are indicated on the left.

DOI: https://doi.org/10.7554/eLife.50034.009

(**Figure 5F and G**). These results indicated that BIRC6 depletion increased the levels of LC3-I without inhibiting the conversion of LC3B-I to LC3B-II, nor the degradation of LC3B-II.

To examine the effect of BIRC6 or UBA6 depletion on the activation of autophagy upon nutrient deprivation, WT, BIRC6-KO and UBA6-KO cells were incubated in serum- and amino-acid-free medium for 0.5, 1 and 2 hr, after which the cells were analyzed by immunoblotting (IB) for LC3B (**Figure 5H and I**, **Figure 5—figure supplement 1E and F**). We observed that in WT cells the level of LC3B-II increased over the first 30 min and then declined for up to 2 hr (**Figure 5H and I**), reflecting the activation and degradation phases of autophagy. In BIRC6-KO or UBA6-KO cells, however, the levels of LC3B-II remained high after 1 and 2 hr of starvation (**Figure 5H and I**, **Figure 5—figure supplement 1E and F**). This effect was likely due to continued replenishment of LC3B-II from the larger pool of LC3B-I in the KO cells. These experiments thus demonstrated that BIRC6 and UBA6 similarly function to reduce LC3B-I levels, limiting the production of LC3B-II during starvation.

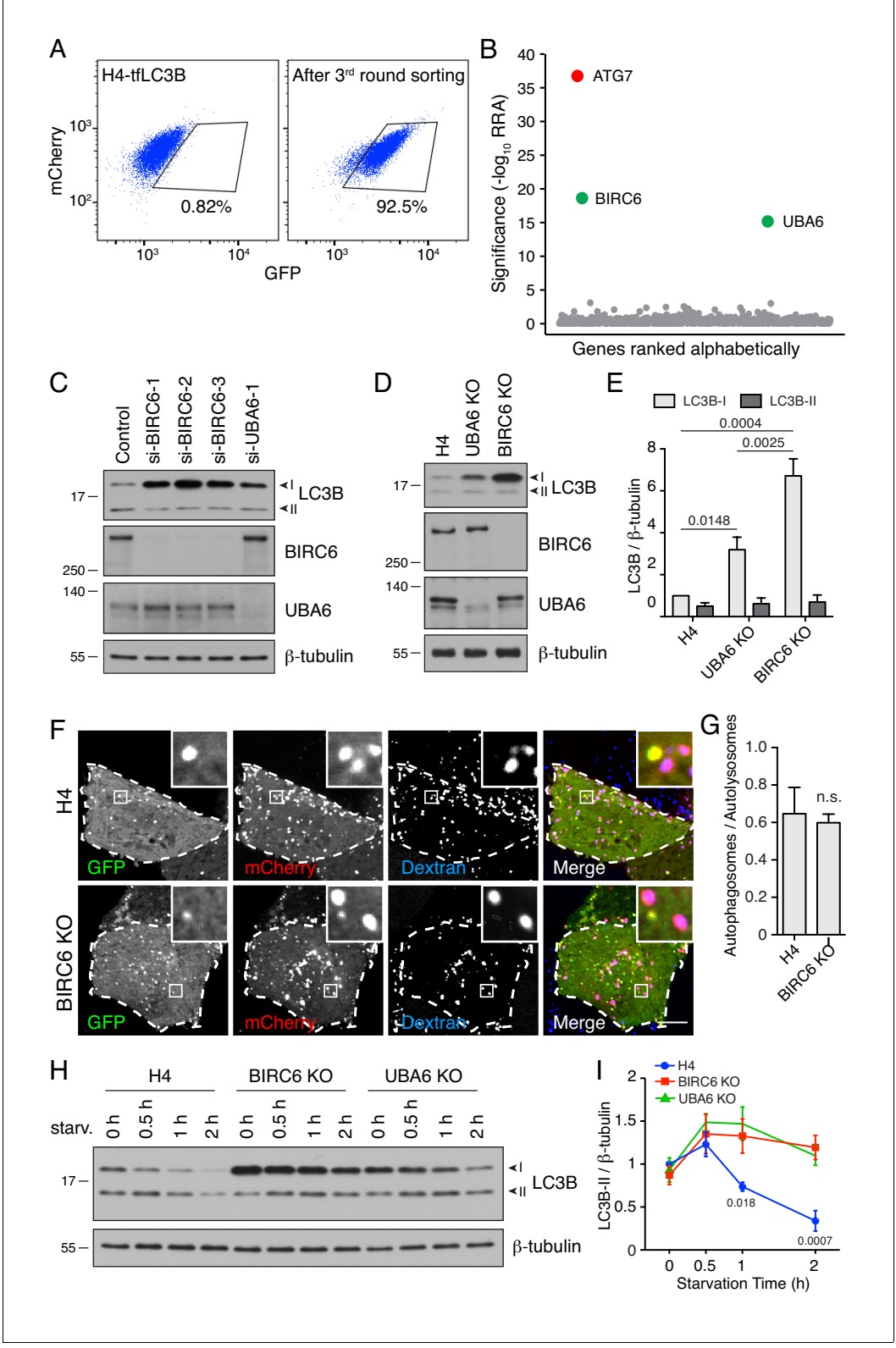

**Figure 5.** BIRC6 depletion increases the level of LC3B-I. (**A**) H4-tfLC3B cells were mutated with a CRISPR/Cas9 lentiviral library targeting ubiquitination-related genes. The figure shows a FACS analysis of the enrichment of library-infected cells with increased GFP fluorescence after three rounds of sorting and expansion. (**B**) Ranking of genes based on RRA. ATG7 (control) is highlighted in red. UBA6 and BIRC6 are highlighted in green. (**C**) H4 cells were transfected with control, BIRC6 (three different siRNAs) or UBA6 siRNAs. Cells were analyzed by SDS-PAGE
*Figure 5 continued on next page*

*Figure 5 continued*

and immunoblotting for LC3B, BIRC6, UBA6 and β-tubulin (loading control). (**D**) SDS-PAGE and immunoblotting of lysates from WT, UBA6-KO and BIRC6-KO H4 cells with antibodies to the proteins on the right. (**E**) Quantification of the ratio of LC3B-I and -II proteins to β-tubulin. The value of LC3B-I/β-tubulin in WT H4 cells was arbitrarily set at 1. Values are the mean ± SEM from three independent experiments such as that shown in D. The indicated *p*-values were calculated using a two-way ANOVA with Tukey's multiple comparisons test. (**F**) WT and BIRC6-KO H4 cells were transfected with a plasmid encoding tfLC3B and allowed to internalize Alexa Fluor 647-conjugated dextran for 16 hr at 37˚C. GFP (green), mCherry (red) and Alexa Fluor 647 (blue) fluorescence was visualized by live-cell imaging. Single-channel images are shown in grayscale. Scale bar: 10 μm. Insets are 4.6x magnifications of the boxed areas. (**G**) The ratio of the number of autophagosomes (red-green–positive puncta) to autolysosomes (red-blue–positive puncta) was determined. Bars represent the mean ± SEM of the ratio in 10 cells from three independent experiments. n.s., not significant, according to an unpaired Student's *t* test. (**H**) WT, BIRC6-KO and UBA6-KO H4 cells were starved of amino acids and serum for the indicated periods, and analyzed by SDS-PAGE and immunoblotting with antibodies to LC3B and β-tubulin. In C, D and H, the positions of molecular mass markers (in kDa) are indicated on the left. (**I**) Ratio of LC3B-II to β-tubulin at different times relative to the ratio in fed WT H4 cells (set at 1). Points represent the mean ± SEM of the ratio from three independent experiments such as that in H. The indicated *p*-values were calculated using a two-way ANOVA with Tukey's multiple comparisons test.

DOI: https://doi.org/10.7554/eLife.50034.010

The following figure supplement is available for figure 5:

**Figure supplement 1.** Screening of a ubiquitination library, increased autophagic flux in UBA6-KO and BIRC6-KO cells, and analysis of LC3B in aggregates, nuclei and microtubules.

DOI: https://doi.org/10.7554/eLife.50034.011

## BIRC6 cooperates with UBA6 to ubiquitinate LC3B

To examine whether BIRC6, like UBA6, was involved in LC3B ubiquitination, we performed the same in vivo assay described above in WT, UBA6-KO and BIRC6-KO cells (*Figure 6A*). We observed that BIRC6 KO reduced LC3B ubiquitination to an even greater extent than UBA6 KO (*Figure 6A*). Moreover, we found that FLAG-BIRC6 expressed by transfection in HEK293T cells was able to pull down both endogenous UBA6 (*Figure 6B*) and transgenic GFP-LC3B (*Figure 6C*), consistent with both UBA6 and BIRC6 interacting to ubiquitinate LC3B. Additional pulldown analyses using truncated FLAG-BIRC6 constructs showed that the segments spanning amino acids 2201–2800, 3301–3800 and 3801–4300 of BIRC6 interacted with GFP-LC3B (*Figure 6—figure supplement 1A and B*). In addition to LC3B, two other LC3-family proteins, LC3A and LC3C, interacted with BIRC6, but the GABARAP-family proteins GABARAP, GABARAPL1, GABARAPL2 did not (*Figure 6—figure supplement 1C*). This finding was consistent with the observation that LC3-family, and not GABARAP-family, proteins were ubiquitinated (*Figure 3G*). We also observed that the K51R LC3B mutant, which was not ubiquitinated (*Figure 4C*), still interacted with BIRC6, albeit to the lesser extent (*Figure 6—figure supplement 1D*). The functional cooperation of UBA6 with BIRC6 was directly tested using an in vitro ubiquitination assay with different combinations of FLAG-BIRC6 immunopurified from transfected HEK293T cells, recombinant HA-Ub, 6xHis-LC3B and GST-UBA6, and a source of ATP (*Figure 6D*). Immunoblot analysis of the complete reaction mixture showed a band corresponding to monoubiquitinated LC3B, which was detected by antibodies to both the HA epitope and LC3B (*Figure 6D*). Omission of any components from the mix abolished this ubiquitination (*Figure 6D*). In addition, substitution of alanine for cysteine-4597 in the active site of the UBC domain of BIRC6 (*Bartke et al., 2004*; *Hao et al., 2004*) also abrogated LC3B monoubiquitination (*Figure 6E*). From these results we concluded that UBA6 and BIRC6 function as E1 and E2/E3 enzymes, respectively, for monoubiquitination of LC3B.

## BIRC6 regulates the accumulation of intracellular aggregates

A critical function of autophagy is the degradation of mutant or otherwise damaged proteins (*Lim and Yue, 2015*). Protein aggregates become ubiquitinated, after which they are recognized by autophagy receptors such as SQSTM1 and NBR1 (*Pankiv et al., 2007*; *Kirkin et al., 2009*). These receptors link the aggregates to LC3 on forming autophagosomes, and the aggregates, receptors and LC3 are all eventually degraded in autolysosomes. To examine the impact of BIRC6 KO on this

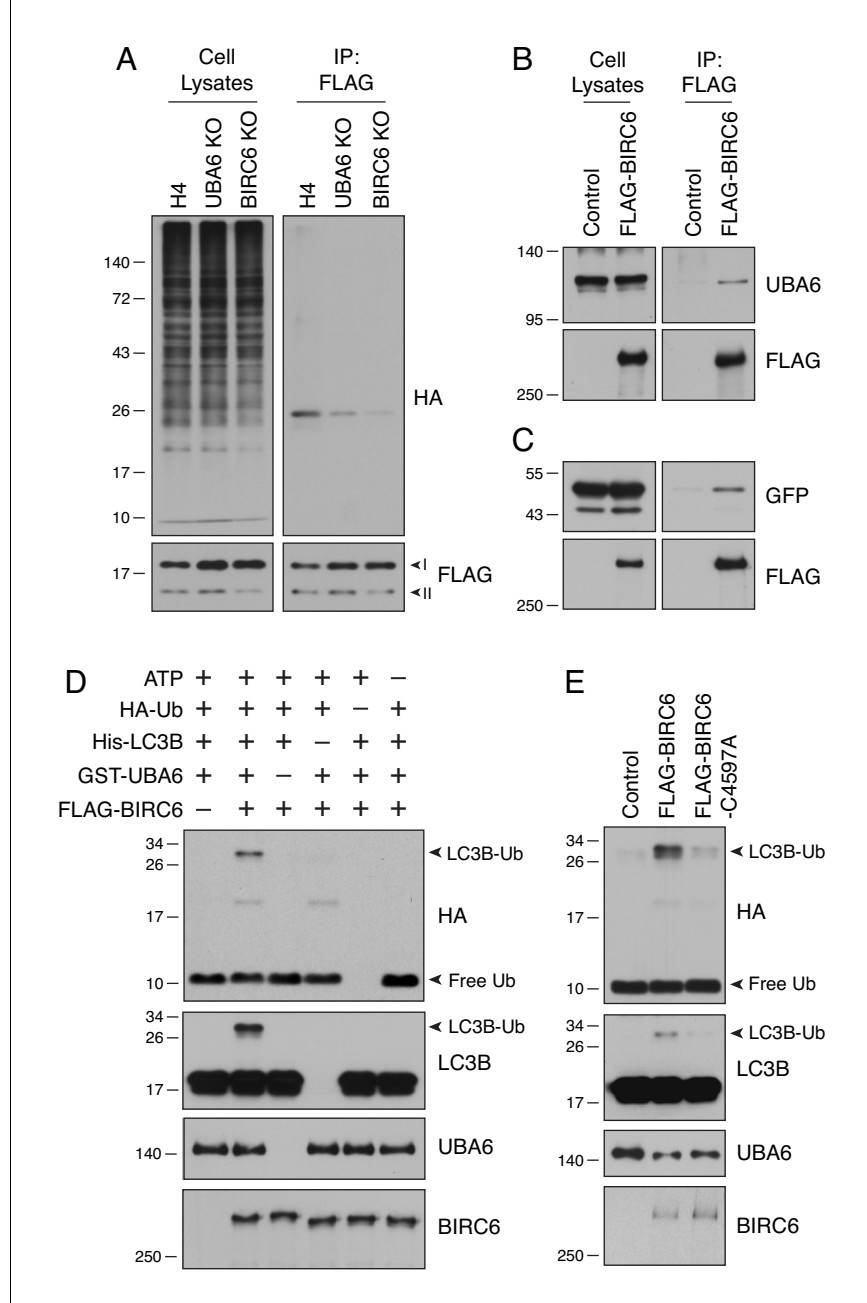

**Figure 6.** UBA6 and BIRC6 function as E1 and E2/E3 enzymes, respectively, for LC3B monoubiquitination. (**A**) WT, UBA6-KO and BIRC6-KO H4 cells were transfected with plasmids encoding HA-Ub and FLAG-LC3B. Cell lysates were analyzed by immunoprecipitation with antibody to the FLAG epitope, followed by SDS-PAGE and immunoblotting with antibodies to the HA and FLAG epitopes. (**B**) WT H4 cells were transfected with control or FLAG-BIRC6 plasmids. Cell lysates were analyzed by immunoprecipitation with antibody to the FLAG epitope, followed by SDS-PAGE and immunoblotting with antibodies to the FLAG epitope and UBA6. (**C**) Plasmids encoding GFP-LC3B and FLAG-BIRC6 were transfected into H4 cells. Cell lysates were analyzed by immunoprecipitation with antibody to the FLAG epitope, followed by SDS-PAGE and immunoblotting with antibodies to GFP and the FLAG epitope. (**D**) FLAG-BIRC6 immunopurified from $10^7$ HEK293T cells transfected with FLAG-BIRC6 plasmid was incubated with recombinant HA-Ub, 6xHis-LC3B and GST-UBA6 in a reaction buffer containing ATP at 37°C for 30 min. Samples were analyzed by SDS-PAGE and immunoblotting with antibodies to the antigens on the right. Notice the ubiquitination of LC3B (LC3B-Ub) only when all the components are present in the mix. (**E**) WT and C4597A-mutant FLAG-BIRC6 were analyzed for their ability to ubiquitinate 6His-LC3B as in D. In all the panels, the positions of molecular mass markers (in kDa) are indicated on the left.

*Figure 6 continued on next page*

*Figure 6 continued*

DOI: https://doi.org/10.7554/eLife.50034.018
The following figure supplement is available for figure 6:

**Figure supplement 1.** Mapping regions of BIRC6 that are required for binding to LC3B, and interaction of BIRC6 with Atg8 family members.
DOI: https://doi.org/10.7554/eLife.50034.019

process, we performed cycloheximide (CHX) chase analysis of the degradation of SQSTM1, NBR1 and LC3B in WT and BIRC6-KO cells (*Figure 7A*). We observed a time-dependent disappearance of LC3B-I in both cell lines, although, as expected, BIRC6-KO cells contained higher levels of LC3B-I at all times after the addition of CHX (*Figure 7A*). Importantly, SQSTM1 and NBR1 exhibited faster rates of degradation in BIRC6-KO relative to WT cells (*Figure 7A–C*). These observations indicated that the larger pool of LC3B-I in BIRC6-KO cells promoted faster degradation of autophagy receptors under conditions of protein synthesis inhibition.

We also examined the effect of BIRC6 KO on the clearance of puromycin-induced ALIS (aggresome-like induced structures), which form by aggregation of prematurely terminated translation products and are degraded by autophagy (*McEwan et al., 2015*; *Szeto et al., 2006*). We observed that addition of puromycin resulted in the emergence of SQSTM1- and Ub-positive ALIS after 2 hr in WT cells, and 8 hr in BIRC6-KO cells (*Figure 7D and E*). Transfection of BIRC6-KO cells with a plasmid encoding FLAG-BIRC6 restored the accumulation of ALIS at 4 hr after puromycin addition (*Figure 7F and G*). Similar observations were made in UBA6 KO cells (*Figure 7—figure supplement 1A and B*). Additionally, we observed that KD of LC3B rescued the accumulation of ALIS in BIRC6-KO cells treated for 4 hr with puromycin (*Figure 7—figure supplement 1C–E*), consistent with the loss of ALIS in BIRC6-KO cells being due to the elevation of LC3B levels. Inhibition of autophagic degradation by incubation with bafilomycin A$_1$ (*Figure 7—figure supplement 1F and G*) or KD of ATG7 (*Figure 7—figure supplement 2A–C*) resulted in increased ALIS accumulation in BIRC6-KO cells, confirming that the clearance of ALIS in these cells was mediated by autophagy.

Uncontrolled aggregate formation underlies the pathogenesis of many diseases, including various neurodegenerative disorders. Parkinson disease (PD), in particular, is associated with formation of aberrant α-synuclein aggregates in neurons (*Polymeropoulos et al., 1997*). To assess a possible effect of BIRC6 depletion on the clearance of α-synuclein aggregates, we transfected rat hippocampal neurons in primary culture with a plasmid encoding the aggregation-prone A53T mutant of α-synuclein (*Conway et al., 1998*), with or without a plasmid encoding an shRNA to rat Birc6. In control cells, we observed the formation of numerous axonal aggregates of α-synuclein (*Figure 8A–C*) that co-localized with endogenous SQSTM1 as well as transgenic HA-Ub and mCherry-LC3B (*Figure 8—figure supplement 1A*), as previously reported (*Watanabe et al., 2012*; *Alexopoulou et al., 2016*). Transfection with the Birc6 shRNA plasmid resulted in a significant reduction in the number of axonal α-synuclein aggregates (*Figure 8A–C*). This reduction could be reversed by co-transfection with shRNA-resistant human BIRC6 (*Figure 8—figure supplement 1B and C*) into the rat Birc6-KD neurons (*Figure 8A–C*) or by incubation of the Birc6-KD neurons with bafilomycin A$_1$ (*Figure 8—figure supplement 2A–C*). Moreover, immunoblot analysis showed a decrease in the amount of α-synuclein in Triton-X-100-insoluble fractions in Birc6-KD neurons relative to control and BIRC6-rescue neurons (*Figure 8D and E*), supporting the conclusion that depletion of Birc6 decreased accumulation of α-synuclein aggregates in rat hippocampal neurons.

## Discussion

The results of our study demonstrate that UBA6 and BIRC6 function as E1 and E2/E3 enzymes, respectively, for the monoubiquitination and subsequent proteasomal degradation of the autophagy protein LC3B. This pathway exerts a negative regulatory effect on autophagy, particularly under stress conditions such as nutrient deprivation (*Figure 5H and I*, *Figure 5—figure supplement 1E and F*), inhibition of protein synthesis (*Figure 7A–C*) and formation of protein aggregates (*Figures 7D, E* and *8A–E*, *Figure 7—figure supplement 1A and B*). Depletion of UBA6 or BIRC6 increases the pool of cytosolic LC3B-I, promoting autophagic flux and clearance of protein aggregates in both non-neuronal and neuronal cells. By dampening LC3 levels, the UBA6-BIRC6 pathway

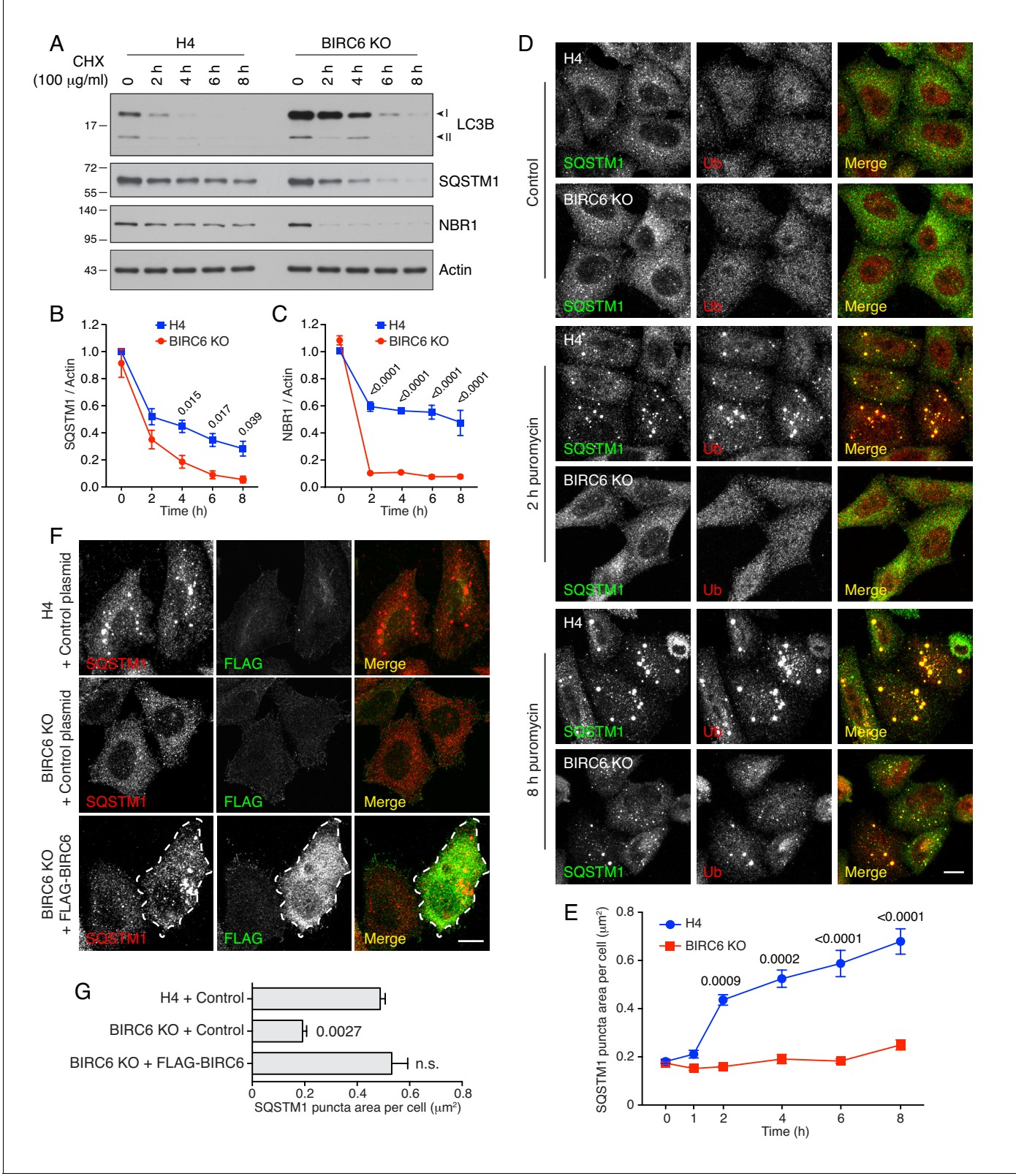

**Figure 7.** BIRC6 KO reduces the accumulation of puromycin-induced ALIS. (**A**) WT and BIRC6-KO H4 cells were incubated with 100 µg/ml cycloheximide (CHX) for the indicated periods. Cells were lysed and analyzed by SDS-PAGE and immunoblotting for the proteins indicated on the right. (**B,C**) Quantification of the ratio of SQSTM1 (**B**) and NBR1 (**C**) to actin at different times after the addition of CHX. Values were normalized to the ratio at time 0 (set at 1). Values are the mean ± SEM from three independent experiments such as that shown in A. (**D**) WT and BIRC6-KO H4 cells were

*Figure 7 continued on next page*

*Figure 7 continued*

incubated without (control) or with 5 µg/ml puromycin for 2 and 8 hr. Cells were subsequently immunostained for SQSTM1 and Ub, and examined by confocal microscopy. Scale bar: 10 µm. (E) Quantification of the area (in µm$^2$) of SQSTM1-positive puncta per cell determined from cells such as those shown in D. Values represent the mean ± SEM of the puncta area in 30 cells from three independent experiments. (F) WT or BIRC6-KO cells were transfected with control plasmid or plasmid encoding FLAG-BIRC6 as indicated in the figure, and incubated with 5 µg/ml puromycin for 4 hr prior to immunostaining with antibodies to SQSTM1 and the FLAG epitope. Scale bar: 10 µm. A FLAG-BIRC6–expressing cell is outlined. In D and F, single-channel images are shown in grayscale. (G) Quantification of the area of SQSTM1 positive puncta per cell. Values represent the mean ± SEM of the puncta area in 30 cells from three independent experiments. The *p*-values shown in panels B, C, E and G were calculated using a one-way ANOVA with Dunnett's multiple comparisons test.

DOI: https://doi.org/10.7554/eLife.50034.012

The following figure supplements are available for figure 7:

**Figure supplement 1.** Analyses of ALIS formation in UBA6-KO and BIRC6-KO cells.

DOI: https://doi.org/10.7554/eLife.50034.013

**Figure supplement 2.** Analyses of ALIS formation in BIRC6-KO cells transfected with ATG7 siRNA.

DOI: https://doi.org/10.7554/eLife.50034.014

may protect cells from excessive autophagy causing autophagy-dependent cell death (*Liu and Levine, 2015*). On the other hand, inhibition of UBA6 or BIRC6 could be used to enhance autophagy for the treatment of neurodegenerative disorders caused by abnormal protein aggregation (*Ross and Poirier, 2004*).

## Comparison to previous CRISPR/Cas9 KO screens

Several recent studies also used genome-wide CRISPR/Cas9 screens to identify novel components and regulators of the autophagy machinery in mammalian cells (*DeJesus et al., 2016*; *Moretti et al., 2018*; *Morita et al., 2018*; *Shoemaker et al., 2019*). As in our study, these screens were performed using cells expressing fluorescently-tagged LC3 or autophagy receptors such as SQSTM1, NPD52, TAX1BP1 or NBR1. All of these screens resulted in the identification of novel candidates, most notably the ER protein TMEM41B, which was shown to participate in autophagosome biogenesis (*Moretti et al., 2018*; *Morita et al., 2018*; *Shoemaker et al., 2019*). Our screen differed from previous ones in that we used cells expressing endogenously-tagged LC3B, in order to avoid artifacts of overexpression. Furthermore, we used a more stringent selection procedure involving isolation of cells that were in the top 1% of GFP intensity in the initial screen, and that maintained that level of fluorescent intensity through three additional rounds of sorting and expansion. This stringency likely accounts for the identification of virtually all known components of the autophagy machinery as top hits in the screen. The relatively high scores of UBA6 and BIRC6 in the whole-genome and ubiquitination screens thus gave us great confidence that these proteins were indeed involved in the regulation of autophagy. It is noteworthy that neither UBA6 nor BIRC6 ranked high in previous screens by other groups. This could be due to the overexpression of LC3B from a CMV promoter (*Shoemaker et al., 2019*), which likely overwhelmed the ability of UBA6 and BIRC6 to target LC3B for degradation. Indeed, we observed that CMV-driven overexpression of FLAG-tagged LC3B resulted in smaller differences in the levels of this protein in WT, UBA6-KO and BIRC6-KO cells (*Figures 3D* and *6A*). Furthermore, UBA6 and BIRC6 could have been missed in previous screens using autophagy receptors as reporters (*DeJesus et al., 2016*; *Moretti et al., 2018*; *Shoemaker et al., 2019*) because the levels of these proteins were unaltered under regular culture conditions, as observed for SQSTM1 and NBR1 in BIRC6-KO cells (*Figure 7A*). Only when cells were treated with cycloheximide did differences in the turnover of these proteins become apparent (*Figure 7A–C*).

## BIRC6-KO does not affect autophagosome-lysosome fusion in our system

A previous shRNA screen of 710 ubiquitination- and autophagy-related genes identified BIRC6 as an autophagy regulator (*Ebner et al., 2018*). In contrast to our findings, however, BIRC6 was reported to promote autophagosome-lysosome fusion through interactions with GABARAP/GABARAPL1 and the SNARE protein syntaxin 17. Moreover, a 1,648-amino-acid N-terminal fragment of BIRC6 was shown to be sufficient for interaction with GABARAP/GABARAPL1 and syntaxin 17, and for

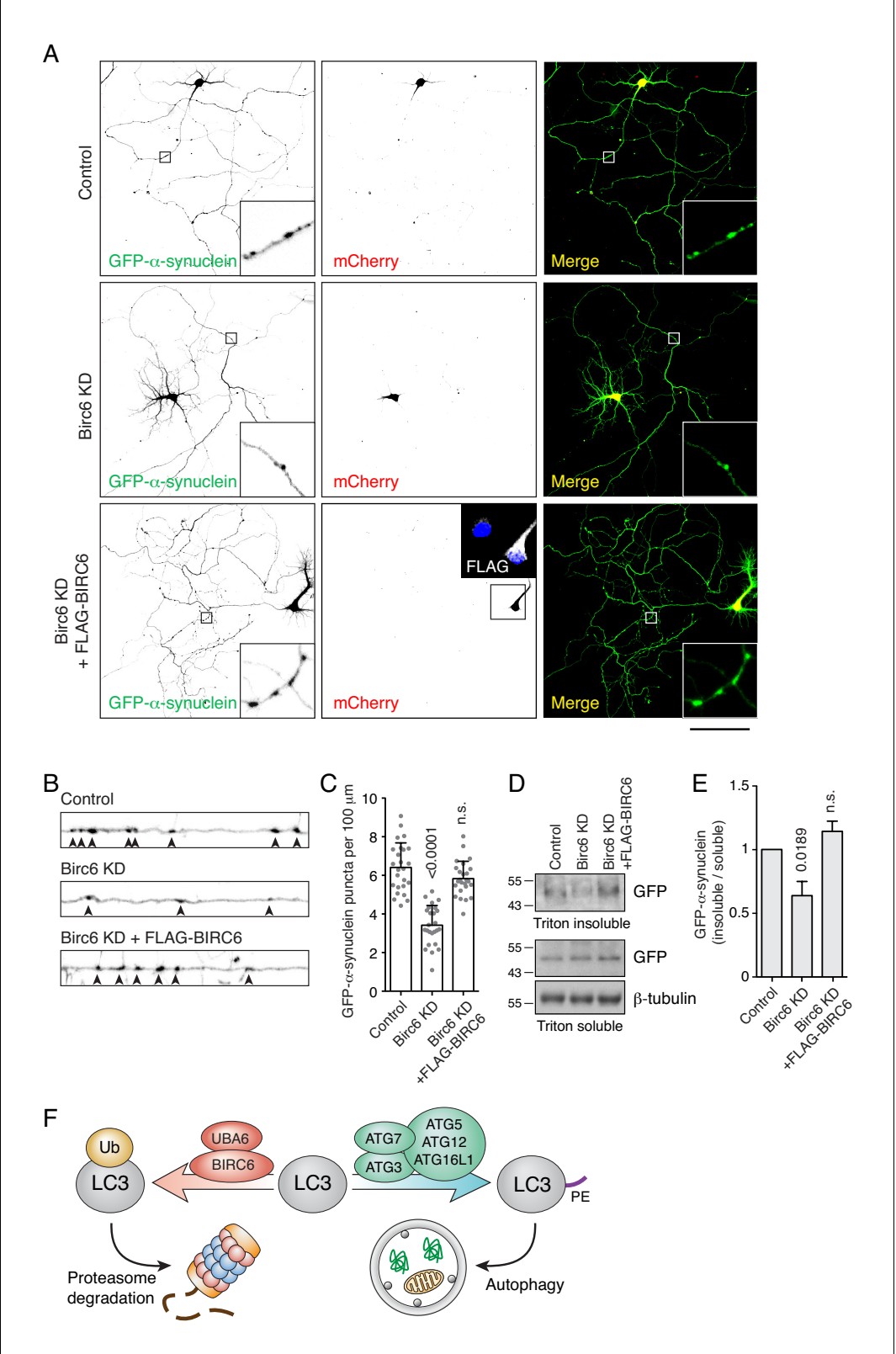

**Figure 8.** Birc6 depletion promotes the clearance of α-synuclein aggregates in neurons. (**A**) Confocal fluorescence microscopy of cultured rat hippocampal neurons co-transfected with plasmids encoding GFP-α-synuclein A53T mutant, mCherry transfection control, rat Birc6 shRNA-mCherry, and/or shRNA-resistant FLAG-tagged human BIRC6 (rescue), as indicated in the figure. Transfections were performed at day-in-vitro 3 (DIV3) and neurons were
*Figure 8 continued on next page*

*Figure 8 continued*

fixed for immunofluorescence microscopy at DIV7. Single-channel images are shown in inverted grayscale. Scale bar: 100 µm. Insets on the left and right columns are 7-fold magnified views from the axons in the boxed area. The inset in the middle bottom row is a 2.2-fold magnified view of FLAG-BIRC6 expression in the boxed area. (B) Magnified and straightened axons from control, Birc6-KD and FLAG-BIRC6-rescue neurons shown in A. Arrowheads indicate α-synuclein aggregates in the axon. (C) Quantification of the number of α-synuclein puncta (i. e., aggregates) per 100 µm of axon. Values are the mean ± SEM from 25 neurons from three independent experiments. The indicated p-values were calculated using a one-way ANOVA with Dunnett's multiple comparisons test. (D) Cultured rat hippocampal neurons were transfected as in A. Neurons were extracted in Triton X-100 buffer and centrifuged. Supernatants were collected as the Triton-soluble fraction, while the pellets were resuspended in 5% SDS buffer as the Triton-insoluble fraction. Fractions were analyzed by SDS-PAGE and immunoblotting with antibodies to GFP (to detect GFP-α-synuclein) and β-tubulin (loading control). The positions of molecular mass markers (in kDa) are indicated on the left. (E) Quantification of the ratio of Triton-insoluble and -soluble GFP-α-synuclein. The value in control-shRNA–transfected neurons was set at 1. Values are the mean ± SEM from three independent experiments such as that in panel D. The indicated p-values were calculated using a one-way ANOVA with Dunnett's multiple comparisons test. (F) Schematic representation of the role of UBA6 and BIRC6 in LC3 ubiquitination and targeting for degradation, decreasing the amount of LC3 that can be modified with PE for its function in autophagy.

DOI: https://doi.org/10.7554/eLife.50034.015

The following figure supplements are available for figure 8:

**Figure supplement 1.** Aggregation of α-synuclein and verification of Birc6 KD in rat hippocampal neurons.
DOI: https://doi.org/10.7554/eLife.50034.016

**Figure supplement 2.** Autophagy dependence of α-synuclein aggregate clearance in rat hippocampal neurons.
DOI: https://doi.org/10.7554/eLife.50034.017

association of BIRC6 to lysosomes (*Ebner et al., 2018*). In our study, we did not observe any defects in autophagosome-lysosome fusion in BIRC6-KO cells (*Figure 5F and G*), nor did we find localization of BIRC6 to lysosomes but rather to the cytosol (*Figure 7F*). Moreover, we found that the C-terminally located UBC domain was essential for the role of BIRC6 in LC3B ubiquitination (*Figure 6E*). We speculate that differences in these studies could be due to the use of different cell types and experimental conditions. In any event, the mechanism involving LC3B ubiquitination and degradation revealed by our study is consistent with the activity of BIRC6 as an E2/E3 enzyme. While our manuscript was in preparation, another study showed that BIRC6 targeted LC3B-I for proteasomal degradation, and that BIRC6 itself was subject to proteasomal degradation promoted by the ubiquitin ligase RNF41/NRDP1 (neuregulin receptor degradation protein-1) and inhibited by CACYBP/SIP (calcyclin-binding protein/siah-1 interacting protein) (*Jiang et al., 2019*). Taken together, these findings reveal the existence of a complex regulatory network for the control of LC3B levels.

## Role of monoubiquitination in LC3B degradation

Our experiments showed that UBA6 and BIRC6 mediate monoubiquitination, rather than polyubiquitination, of LC3B on lysine-51 (*Figure 4C*). The fact that this modification targets LC3B-I for proteasomal degradation is remarkable in light of early notions that monoubiquitination played mainly regulatory, non-degradative roles (*Haglund et al., 2003*), whereas polyubiquitination was the signal for degradation (*Thrower et al., 2000*). More recent evidence, however, suggests that monoubiquitination can also target proteins for proteasomal degradation (*Boutet et al., 2007*; *Dimova et al., 2012*), especially for small proteins of 20–150 residues (*Shabek et al., 2012*). Moreover, a systematic analysis revealed that approximately half of all the proteins degraded by proteasomes in human cells are monoubiquitinated or multi-monoubiquitinated (*Braten et al., 2016*). These findings are consistent with our conclusion that monoubiquitination of the 125-amino-acid LC3B is the signal that targets it for degradation.

## Insights into the role of UBA6 in ubiquitination

UBA6 is one of eight E1 enzymes involved in the conjugation of 17 Ub-like proteins encoded in the human genome (*Schulman and Harper, 2009*). In particular, UBA6 has been reported to activate both Ub and the Ub-like protein FAT10 (*Pelzer et al., 2007*; *Chiu et al., 2007*; *Jin et al., 2007*). UBA6-mediated FAT10ylation has also been implicated in targeting proteins for proteasomal

degradation (*Aichem et al., 2012*) as well as in the intracellular defense against bacterial infection (*Spinnenhirn et al., 2014*). Except for one study (*Lee et al., 2011*), however, little has been shown about the role of UBA6-mediated ubiquitination. Herein we show that UBA6 uses BIRC6 as an E2/E3 to ubiquitinate LC3. Nevertheless, small amounts of monoubiquitinated LC3B can still be found in UBA6-KO and BIRC6-KO cells (*Figure 6A*). This finding suggests that other ubiquitinating enzymes could partially compensate for the absence of UBA6 or BIRC6. These alternative enzymes, however, likely play minor roles in LC3B ubiquitination, because they were not identified in our screens.

## Impact of increased LC3 levels on autophagy and aggregate clearance

Although most of our work dealt with LC3B, the homologous LC3A and LC3C were also ubiquitinated (*Figure 3G*), and UBA6 KO increased LC3A-I in addition to LC3B-I (*Figure 2—figure supplement 1D*). In the same assays, GABARAP, GABARAPL1 and GABARAPL2 were not ubiquitinated (*Figure 3G*) and the levels of GABARAP and GABARAPL1 did not change in UBA6-KO cells (*Figure 2—figure supplement 1D*). Moreover, BIRC6 interacted with LC3A, LC3B and LC3C more strongly than with GABARAP, GABARAPL1 and GABARAPL2 (*Figure 6—figure supplement 1C*). UBA6-BIRC6–mediated ubiquitination and degradation thus appear to be restricted to the LC3 sub-group of the Atg8 family. Recent studies showed that both LC3 and GABARAP family members contribute to autophagic degradation of protein aggregates and mitochondria (*Vaites et al., 2017*; *Nguyen et al., 2016*), although GABARAPs seemed more important (*Vaites et al., 2017*). We find that increasing the levels of LC3B by BIRC6-KO in H4 neuroglioma cells reduces the half-life of SQSTM1 and NBR1 (*Figure 7A–C*). Moreover, depletion of BIRC6 or UBA6 decreases the formation of SQSTM1- and Ub-decorated protein aggregates in both H4 cells (*Figure 7D–G*, *Figure 7—figure supplement 1A–G*, *Figure 7—figure supplement 2A–C*) and rat hippocampal neurons (*Figure 8A–E*, *Figure 8—figure supplement 1A–C*, *Figure 8—figure supplement 2A–C*). Therefore, at least in these brain-derived cells, decreasing the levels of LC3 enhances autophagic flux and decreases accumulation of protein aggregates. Our findings are in line with a previous report that overexpression of LC3B in a transgenic mouse model of Alzheimer disease inhibited amyloid-β-peptide-induced neuron degeneration by enhancing autophagy flux (*Hung et al., 2015*). In addition, another study showed that overexpression of a LC3B transgene attenuated lung injury in a mouse model of sepsis, probably by enhancement of autophagosome clearance (*Lo et al., 2013*). These findings not only support the notion that elevated LC3 levels promote autophagy flux, but also suggest that increasing LC3 levels could have therapeutic applications.

## Potential use of UBA6 or BIRC6 inhibition to treat protein aggregation diseases

Intracellular accumulation of pathogenic protein aggregates is a common cause of various neurodegenerative disorders (*Ross and Poirier, 2004*). This accumulation usually results from overproduction of normal proteins or from synthesis of mutant proteins that overwhelm the capacity of the autophagy machinery to clear them. An example of such disorders is PD, a neurodegenerative disease characterized by the presence of Lewy bodies containing aggregated α-synuclein (*Polymeropoulos et al., 1997*). Since α-synuclein aggregates are mainly degraded by autophagy (*Webb et al., 2003*), autophagy enhancement has been proposed as a possible therapy for PD (*Williams et al., 2006*). Indeed, autophagy enhancing agents targeting AMPK (*Patil et al., 2014*), mTORC1 (*Crews et al., 2010*; *Bai et al., 2015*), BECN1 (*Savolainen et al., 2014*) and lysosomal glucocerebrosidase (*Richter et al., 2014*) have all been shown to increase α-synuclein clearance and to have a neuroprotective effect in animal models of PD. However, targeting AMPK and mTORC1 is problematic because, as central nodes of nutrient and energy signaling pathways, they regulate numerous of cellular processes, including cell growth, protein synthesis and metabolism (*Saxton and Sabatini, 2017*; *Mihaylova and Shaw, 2011*). The decreased accumulation of α-synuclein aggregates by depletion of UBA6 or BIRC6 shown here suggests that these proteins could also be targeted for pharmacologic treatment of PD and other protein aggregation disorders.

# Materials and methods

## Key resources table

| Reagent type (species) or resource | Designation | Source or reference | Identifiers | Additional information |
|---|---|---|---|---|
| Antibody | LC3B (rabbit, monoclonal) | Cell Signaling Technology | 3868 | IB: 1:2000 |
| Antibody | UBA6 (rabbit polyclonal) | Cell Signaling Technology | 13386 | IB: 1:1000 |
| Antibody | β-tubulin (rabbit polyclonal) | Cell Signaling Technology | 2146 | IB: 1:1000 |
| Antibody | MYC epitope (rabbit polyclonal) | Santa Cruz Biotechnology | sc-789 | IB: 1:5000 |
| Antibody | Ub (mouse monoclonal) | Thermo Fisher Scientific | 13–1600 | IB: 1:1000 |
| Antibody | Ub (mouse monoclonal) | Enzo Life Sciences | BML-PW8805-0500 | IF: 1:500 |
| Antibody | HA epitope (mouse monoclonal) | BioLegend | 901501 | IB: 1:2000 |
| Antibody | FLAG epitope (mouse monoclonal) | Sigma-Aldrich | F1804 | IB: 1:1000 |
| Antibody | SQSTM1 (mouse monoclonal) | BD Biosciences | 610833 | IB: 1:1000 |
| Antibody | SQSTM1 (rabbit polyclonal) | Enzo Life Sciences | BML-PW9860-0100 | IF: 1:500 |
| Antibody | NBR1 (rabbit monoclonal) | Cell Signaling Technology | 9891 | IB: 1:1000 |
| Antibody | BIRC6 (rabbit monoclonal) | Cell Signaling Technology | 8756 | IB: 1:1000 |
| Antibody | GFP (rabbit polyclonal) | Thermo Fisher Scientific | A11122 | IB: 1:1000 |
| Antibody | Actin (mouse monoclonal) | Cell Signaling Technology | 3700 | IB: 1:1000 |
| Chemical compound, drug | Bafilomycin A$_1$ | Sigma-Aldrich | B1793 | |
| Chemical compound, drug | MG132 | Sigma-Aldrich | M7449 | |
| Chemical compound, drug | Cycloheximide | Sigma-Aldrich | C4859 | |
| Chemical compound, drug | Puromycin | Sigma-Aldrich | P8833 | |
| Chemical compound, drug | Polybrene | Sigma-Aldrich | H9268 | |
| Chemical compound, drug | MgATP | Boston Biochem | B-20 | |
| Chemical compound, drug | Fibronectin | Sigma-Aldrich | F2006 | |
| Cell line (*Homo sapiens*) | H4 | ATCC | HTB-148 | |
| Cell line (*Homo sapiens*) | HeLa | ATCC | CCL-2 | |
| Cell line (*Homo sapiens*) | HEK293T | ATCC | CRL-11268 | |
| Recombinant DNA | HA-Ub | Addgene; gift from Edward Yeh | 18712 | |
| Recombinant DNA | GFP-α-synuclein A53T | Addgene; gift from David Rubinsztein | 40823 | |

*Continued on next page*

*Continued*

| Reagent type (species) or resource | Designation | Source or reference | Identifiers | Additional information |
|---|---|---|---|---|
| Recombinant DNA | GFP-LC3B | *Jia et al., 2017* | | |
| Recombinant DNA | GFP-mCherry-LC3B | *Jia et al., 2017* | | |
| Recombinant DNA | pSpCas9 (BB)—2A-GFP | Addgene; gift from Feng Zhang | 40823 | |
| Recombinant DNA | FLAG-BIRC6 | Gift from Mikihiko Naito | | Kawasaki Hospital, Kobe, Japan |
| Recombinant DNA | pMD2.G | Addgene; gift from Didier Trono | 12259 | |
| Recombinant DNA | psPAX | Addgene: gift from Didier Trono | 12260 | |
| Recombinant DNA | lentiCRISPR v2 | Addgene; gift from Feng Zhang | 52961 | |
| Sequence-based reagent | UBA6 siRNA | Thermo Fisher Scientific | 4392420-s30516 | |
| Sequence-based reagent | UBA6 siRNA | Thermo Fisher Scientific | 4392420-s30517 | |
| Sequence-based reagent | BIRC6 siRNA | Thermo Fisher Scientific | 4427037-s33037 | |
| Sequence-based reagent | BIRC6 siRNA | Thermo Fisher Scientific | 4427037-s33038 | |
| Sequence-based reagent | BIRC6 siRNA | Thermo Fisher Scientific | 4427037-s33039 | |
| Sequence-based reagent | Non-targeting siRNA | Eurofins Scientific | | UGGUUUACAUGUCGACUAAUUU |
| Peptide, recombinant protein | GST-UBA6 | Boston Biochem | E-307 | |
| Peptide, recombinant protein | 6xHis-LC3B | ProSpec | PRO-076 | |
| Peptide, recombinant protein | HA-Ub | Boston Biochem | U-110 | |
| Commercial assay or kit | Q5 Site-Directed Mutagenesis Kit | New England Biolabs | E0552 | |
| Commercial assay or kit | Gibson Assembly Master Mix | New England Biolabs | E2611 | |
| Commercial assay or kit | Human CRISPR KO Pooled Library (GeCKO v2) | Addgene; gift from Feng Zhang | 1000000048 | |
| Commercial assay or kit | Endura Electro Competent cells | Lucigen | 60242 | |
| Software, algorithm | Prism 7 | GraphPad Software | | https://www.graphpad.com |
| Software, algorithm | MAGeCK | | | https://sourceforge.net/p/mageck/wiki/Home |
| Software, algorithm | ImageJ | National Institutes of Health | | https://imagej.nih.gov/ij/ |

IB: Immunoblotting; IF immunofluorescence

## Cell culture and transfection

H4, H4-derived-KO, HeLa and HEK293T cells were grown in Dulbecco's modified Eagle's medium (DMEM, Corning, 15–013-CV) with 10% fetal bovine serum (FBS, Corning, 35–011-CV), 100 IU/ml

penicillin, 100 µg/ml streptomycin (Corning, 30–002 CI) and 2 mM L-glutamine (Corning, 25–005 CI) at 37°C, 5% $CO_2$. Cells were confirmed to be free of mycoplasma contamination. Transfection of siRNAs was performed with Oligofectamine (Thermo Fisher Scientific, 12252011), and transfection of plasmids was performed with Lipofectamine 2000 (Thermo Fisher Scientific, 11668019), both according to the manufacturers' instructions. Starvation was performed by incubating cells with DMEM without amino acids and serum (MyBioSource, MBS653087).

## Plasmids

FLAG-LC3A, FLAG-LC3B, FLAG-LC3C, FLAG-GABARAP, FLAG-GABARAPL1, FLAG-GABARAPL2 were generated by subcloning coding sequences for each protein into pCI-neo vector (Promega, E1841) with an N-terminal FLAG tag. FLAG-LC3B-G120A, FLAG-LC3B-K5R-K8R, FLAG-LC3B-K30R, FLAG-LC3B-K39R-K42R, FLAG-LC3B-K49R-K51R, FLAG-LC3B-K65R, FLAG-LC3B-K103R, FLAG-LC3B-K122R, FLAG-LC3B-K49R, FLAG-LC3B-K51R and FLAG-BIRC6-C4597A were generated using the Q5 Site-Directed Mutagenesis Kit. The pSuper-Birc6-shRNA plasmid was generated by cloning an shRNA for rat Birc6 (5'-CACCCCAGTAGTCATACCA-3') into pSuper.neo+mCherry (*Guo et al., 2016*).

## Genome editing by CRISPR/Cas9

To generate cells expressing endogenously tagged GFP-mCherry-LC3B (H4-tfLC3B), the targeting sequence for LC3B (TTCTCCGACGGCATGGTGCA) was cloned into the pSpCas9 (BB)−2A-GFP plasmid. Donor DNA for homology recombination consisted of a 1000 bp left homology arm, DNA sequence encoding GFP-mCherry, and 1000 bp right homology arm (*Figure 1—figure supplement 1A*). H4 cells were co-transfected with CRISPR/Cas9 plasmid and donor DNA. GFP-positive cells were sorted on a FACS Aria II Flow Cytometer (BD Biosciences) after 24 hr, and propagated. GFP-mCherry double-positive cells were collected by FACS one week after transfection. Single-cell clones were isolated by serial dilution on 96-well plates. The insertion of the GFP-mCherry fragment was confirmed by DNA sequencing and immunoblotting (*Figure 1—figure supplement 1B*).

UBA6-KO and BIRC6-KO H4 cells were generated by CRISPR/Cas9 as previously described (*Pu et al., 2015*). The targeting sequences for UBA6 (CGAGCCTGTGGCCGCCCATC and CTCCGG TCGAGAGCGAGTTC) and BIRC6 (GCATGCACTGCGACGCCGAC and TCTCGCTTCCCCGAG TCGCG) were cloned separately into pSpCas9 (BB)−2A-GFP plasmid. H4 cells were co-transfected with two plasmids containing the different targeting sequences for the same gene, and GFP-positive cells were selected on a FACS Aria II Flow Cytometer after 24 hr. Single-cell clones were isolated on 96-well plates. After 14 days, genomic DNA was extracted, and the cleavage of the target sequence tested by PCR. The KO was confirmed by DNA sequencing and immunoblotting.

## Genome-wide CRISPR/Cas9 screen

A human CRISPR KO pooled library (GeCKO v2) was used to introduce mutations in the H4-tfLC3B genome. HEK293T cells were co-transfected with the GeCKO v2 library, pMD2.G and psPAX. Supernatants were collected 48 hr after transfection, filtered on a 0.2 µm filter unit, and stored at −80°C. To test the viral titer, 0.5 million H4-tfLC3B cells were seeded on each well of a 6-well plate 24 hr prior to infection. The cells were incubated with lentiviral supernatants with 5 µg/ml polybrene for 4 hr, and one extra well of cells was incubated with regular medium without viral supernatants as control. Viral supernatants were then discarded, and fresh medium was added. The following day, cells were trypsinized and 10,000 cells were seeded into each well of a 24-well plate in duplicate. One day later, one well was supplemented with 1 µg/ml puromycin, while the other well was maintained in regular medium. When puromycin killed all the cells in the no-virus condition, the number of cells in each well was counted. The multiplicity of infection (MOI) was calculated by dividing the number of cells in puromycin-selected wells by that in the unselected wells.

For the large-scale screen, 200 million H4-tfLC3B cells were infected with the lentiviral pool at a MOI of 0.3. The transduced cells were selected by adding 1 µg/ml puromycin 24 hr after infection. After puromycin selection for 7 days, the initial screen was performed by collecting the top 1% cells with increased GFP signal by sorting on a FACS Aria II Flow Cytometer. The sorted cells were propagated to 10 million and subjected to the next round of sorting with the same gating.

## Next-generation sequencing

Genomic DNA from 60 million unsorted cells (500 × coverage of the GeCKO v2 library) and 6 million sorted cells was isolated using Blood and Cell Culture DNA Midi Kit (QIAGEN, 13343) according to the manufacturer's instructions. DNA fragments encoding sgRNAs were amplified by PCR from 250 µg (for unsorted cells) or 25 µg (for sorted cells) genomic DNA using the primers: AATGGACTATCA TATGCTTACCGTAACTTGAAAGTATTTCG/CAAAAAAGCACCGACTCGGTGCCACTTTTTCAAG. A second PCR was then performed to attach Illumina overhang adapters using the primer set TCG TCGGCAGCGTCAGATGTGTATAAGAGACAGGCTTACCGTAACTTGAAAGTATTTCG/ GTCTCGTGGGCTCGGAGATGTGTATAAGAGACAGACCGACTCGGTGCCACTTTTTCAAG. Indices and Illumina sequencing adapters were added using the Nextera XT Index kit following the manufacturer's protocol (Illumina). Libraries were multiplexed and sequenced on an Illumina MiSeq using v2 chemistry, generating an average of 2.5 million reads (2 × 150 bp) per sample. The sgRNA sequences that were 20 bp in length and that completely matched the GeCKO v2 library were selected, and the number of reads of each sgRNA was calculated and analyzed using the MAGeCK algorithm.

## Secondary and ubiquitination screen

The pooled secondary and ubiquitination-related CRISPR libraries were constructed based on the sgRNA sequences from two published whole-genome CRISPR screens (*Joung et al., 2017*; *Wang et al., 2015*). In the secondary library, 432 top-ranked genes from the primary screen were targeted, while in the ubiquitination library, 661 major E1, E2, E3 and deubiquitinating enzymes were targeted. Both libraries contained 1000 nontargeting sgRNAs as controls. Oligonucleotides containing the guide sequences were synthesized by Twist Bioscience as a pool, and amplified by PCR using the primers Oligo-Fwd: GTAACTTGAAAGTATTTCGATTTCTT GGCTTTATATATCTTG TGGAAAGGAC GAAACACC and Oligo-Knockout-Rev: ACTTTTTCAAGTTGATAACGGACTAG CC TTATTTTAACTTGCTATTTCTAGCT CTAAAAC. PCR products were ligated into lentiCRISPR v2 using Gibson Assembly Master Mix. The ligation products were purified and concentrated by isopropanol precipitation, then transformed into Endura ElectroCompetent cells according to manufacturer's directions. The number of colonies were counted to ensure there were more than 500 colonies per sgRNA in the library. The colonies were collected and subjected to plasmid preparation. The production of the lentiviral pool and screens in H4-tfLC3 cells were performed as described for the primary screen.

## Flow cytofluorometry

After starvation, bafilomycin A$_1$ incubation or library infection, cells were detached with 0.5% trypsin and 5 mM EDTA, pelleted and resuspended in regular DMEM with 10% FBS. Fluorescence intensities were measured on an LSRFortessa Flow Cytometer (BD Biosciences), and the cells with increased GFP signal were collected on a FACS Aria II Flow Cytometer. Scatter plots were generated with FlowJo.

## Immunofluorescence microscopy and live-cell imaging

Cells were grown on glass coverslips coated with 5 µg/ml fibronectin for 24 hr prior to each experiment. Cells were washed once with PBS, fixed in 4% paraformaldehyde (PFA) in PBS for 20 min, and permeabilized in 0.2% Triton X-100 (Sigma-Aldrich, T9284) for 20 min. Cells were incubated with primary antibodies diluted in 0.2% BSA (Sigma-Aldrich, A7030) in PBS for 1 hr at 37°C, washed three times with PBS, and incubated with Alexa Fluor-conjugated secondary antibodies (Thermo Fisher Scientific) for 30 min at 37°C. Cells were washed with PBS and mounted with DAPI-Fluoromount-G (Electron Microscopy Sciences, 17984–24).

Neurons were washed once with PBS-CM (PBS supplemented with 0.1 mM CaCl$_2$ and 1 mM MgCl$_2$), fixed with 4% PFA and 4% sucrose in PBS-CM, and permeabilized with 0.2% Triton X-100 in PBS-CM. Neurons were blocked with 0.2% gelatin in PBS-CM and incubated with primary antibodies diluted in 0.2% gelatin for 1 hr at 37°C, then washed three times with PBS-CM and incubated with secondary antibodies. After three more washes with PBS-CM and one wash with water, neurons were mounted with DAPI-Fluoromount-G.

Live-cell imaging was performed in four-well Nunc Lab-Tek Chambered Coverglasses (Thermo Fisher Scientific, 155383) coated with 5 µg/ml fibronectin. To inhibit lysosomal degradation, cells

were incubated with 50 nM bafilomycin A$_1$ for 2 hr prior to imaging. Labeling of lysosomes was performed by loading cells with Alexa Fluor 647-dextran in complete DMEM medium at 37°C for 16 hr after transfection. Fluorescence was visualized on a Zeiss LSM780 confocal microscope (Carl Zeiss). Image analysis was performed with ImageJ.

## Immunoprecipitation and immunoblotting

H4 cells were grown to ~90% confluency, washed with PBS, harvested by scraping, and lysed on ice for 20 min in lysis buffer (150 mM NaCl, 50 mM Tris-HCl pH 7.4, 5 mM EDTA, 1% Triton X-100, 3% glycerol [Sigma-Aldrich, G6279]) with a protease inhibitor cocktail (Roche, 11697498001). Cytosolic extracts were cleared by centrifugation at 13,000 g for 20 min at 4°C and incubated with anti-FLAG (Sigma-Aldrich, F4799) or anti-HA (Thermo Fisher Scientific, 88836) magnetic beads at 4°C for 4 hr. After washing three times in lysis buffer, proteins were eluted from beads by incubation with 3xFLAG peptide (Thermo Fisher Scientific, A36798) or HA peptide (Thermo Fisher Scientific, 26184).

## In vitro ubiquitination assay

For the in vitro ubiquitination assay, FLAG-tagged BIRC6 was immunoisolated as described (Bartke et al., 2004). Briefly, HEK293T cells expressing FLAG-tagged BIRC6 were lysed on ice for 20 min in digitonin lysis buffer (20 mM HEPES pH 7.5, 10 mM KCl, 1.5 mM MgCl$_2$, 1 mM EDTA, 0.05% digitonin [Sigma-Aldrich, D141], 250 mM sucrose) with a protease inhibitor cocktail. Cell lysates were cleared by centrifugation at 20,000 g for 15 min at 4°C, and NaCl was added to 150 mM final concentration. Cleared lysates were incubated with anti-FLAG magnetic beads at 4°C for 4 hr. Beads were washed with 1% Triton X-100 in PBS followed by 0.1% Triton X-100 in PBS, and by PBS.

Immunopurified FLAG-BIRC6 on beads was incubated with a reaction mixture containing 125 nM GST-UBA6, 6 µM 6xHis-LC3B, 10 µM HA-Ub, and 10 µM MgATP, in a final volume of 30 µl of E3 ligase buffer (Boston Biochem, B-71) at 37°C for 1 hr. Reactions were terminated by mixing supernatants with 4xLDS (lithium dodecyl sulfate) sample buffer (Thermo Fisher Scientific, NP0007). FLAG-BIRC6 was eluted from the beads by incubation with 3xFLAG peptide. Samples were analyzed by immunoblotting.

## Preparation, culture and transfection of neurons

Primary rat hippocampal neurons were prepared from Sprague-Dawley rats at embryonic stage E18 as previously described (Farías et al., 2012). Hippocampi were dissociated with 0.25% trypsin in 2.2 mM EDTA and plated onto polylysine-coated plates. Neurons were maintained in DMEM supplemented with 10% horse serum for 3 hr, then medium was changed to Neurobasal medium (Thermo Fisher Scientific, 21103049), supplemented with serum-free B27 (Thermo Fisher Scientific, A3582801), 100 U/ml penicillin and 100 µg/ml streptomycin. Neurons were transfected with the indicated plasmids using Lipofectamine 2000 at day-in-vitro 3 (DIV3) and analyzed at DIV7 by immunofluorescence microscopy or subcellular fractionation.

## Extraction of soluble and insoluble α-synuclein

Cultured primary hippocampal neurons were lysed on ice for 15 min in Triton buffer (50 mM Tris-HCl pH 7.5, 150 mM NaCl, 1% Triton X-100, and 1 mM EDTA supplemented with protease inhibitor cocktail) and centrifuged at 15,000 g for 10 min at 4°C. The supernatant was collected as the Triton-soluble fraction. After washing three times with Triton buffer, the pellet was resuspended in SDS buffer (Triton buffer supplemented with 5% SDS [Sigma-Aldrich, L3771]), heated at 95°C for 10 min, and then centrifuged at 15,000 g for 10 min at room temperature. The supernatant was collected as the Triton-insoluble fraction. Triton-soluble and -insoluble fractions were mixed with LDS sample buffer and analyzed by immunoblotting.

## Antibodies used in supplementary experiments

Antibodies to the following antigens (supplier and catalog information in parentheses) were used in the supplementary experiments: LC3A (Cell Signaling Technology, 4599), GABARAP (Cell Signaling Technology, 13733), GABARAPL1 (Cell Signaling Technology, 26632), ATG7 (Cell Signaling Technology, 8558), ATG3 (Cell Signaling Technology, 3415), ATG16L1 (Cell Signaling Technology, 8089), ATG5 (Cell Signaling Technology, 12994), ATG12 (Cell Signaling Technology, 4180), WIPI2 (Bio-Rad,

MCA5780), p-BECN1 (Abbiotec, 254515), BECN1 (Cell Signaling Technology, 3495), ATG14 (MBL International, PD026), p-ATG13 (Rockland Immunochemicals, 600–401 C49S), ATG13 (Cell Signaling Technology, 13468), p-ULK1 (Cell Signaling Technology, 12753), ULK1 (Cell Signaling Technology, 8054), p-S6K (Cell Signaling Technology, 9234), S6K (Cell Signaling Technology, 2708), p-4EBP (Cell Signaling Technology, 2855), 4EBP (Cell Signaling Technology, 9452), p-TSC2 (Cell Signaling Technology, 3617), TSC2 (Cell Signaling Technology, 4308), p-AKT (Cell Signaling Technology, 13038), AKT (Cell Signaling Technology, 4691), ATG9A (Cell Signaling Technology, 9730), TGN46 (Bio-Rad, AHP500GT), Calnexin (EMD Millipore, MAB3126), GM130 (BD Biosciences, 610823), LAMTOR4 (Cell Signaling Technology, 12284), Rab5 (Cell Signaling Technology, 2143), APPL1 (Santa Cruz Biotechnology, sc-271901), EEA1 (Cell Signaling Technology, 3288), Alexa Fluor 647 Phalloidin (Thermo-Fisher Scientific, A22287), Lamin A/C (Cell Signaling Technology, 4777), KIF5B (Abcam, ab167429).

## Real-time quantitative PCR (RT-qPCR)

WT, UBA6-KO and BIRC6-KO cells were grown to ~90% confluency. Total RNA was extracted with the RNeasy Mini Kit (Qiagen, 74104). Reverse transcription was carried out using SuperScript IV VILO Master Mix (ThermoFisher Scientific, 11756050). Gene expression was measured using PowerUp SYBR Green Master Mix (ThermoFisher Scientific, A25741). RT-qPCR was performed using the AriaMx Real-Time PCR System (Agilent Technologies). Gene expression was normalized to GAPDH gene expression. Primers used for RT-qPCR were: AAGCAGCTTCCTGTTCTGGAT (forward)/GATTGGTGTGGAGACGCTGA (reverse) for LC3B; CTGACTTCAACAGCGACACC (forward)/TAGCCAAATTCGTTGTCATACC (reverse) for GAPDH.

## Nuclear fractionation

The procedure for separation into nuclear and cytoplasmic fractions was adapted from a previous study examining the effect of acetylation on LC3 function (*Huang et al., 2015*). Briefly, WT and BIRC6-KO cells were washed twice with ice-cold PBS and scraped into hypotonic lysis buffer (1 mM EGTA, 1 mM EDTA, 2 mM MgCl$_2$, 10 mM KCl, 1 mM dithiothreitol, pH 7.2) containing a protease inhibitor cocktail (Roche, 11697498001). The cell suspensions were incubated on ice for 30 min to allow cell lysis. The lysis efficiency was examined by microscopy, to ensure disruption of the cells. The lysates were centrifuged at 750 g for 5 min to pellet nuclei. The supernatants were collected as the cytoplasmic fraction. The pellets were washed twice in hypotonic lysis buffer and resuspended in the same buffer. The supernatants and pellets were mixed with 4xLDS sample buffer and analyzed by SDS-PAGE and immunoblotting.

## Microtubule co-sedimentation

Microtubule co-sedimentation was performed as described previously, with minor modifications (*Tonami et al., 2007*). Briefly, WT and BIRC6-KO cells were washed with cold PBS, harvested by scraping, and lysed in PIPES buffer (80 mM PIPES pH6.8, 1 mM MgCl$_2$, 1 mM EGTA, 100 mM NaCl, 1% Triton X-100) containing a protease inhibitor cocktail (Roche, 11697498001). Lysates were centrifuged at 4°C for 20 min. Supernatants were supplemented with 1 mM GTP (final concentration) (ThermoFisher Scientific, R1461) and 40 µM Taxol (final concentration) (ThermoFisher Scientific, P3456), and incubated at 37°C for 60 min to allow the polymerization of microtubules. Microtubules were collected by centrifugation at 20,000 g for 15 min at room temperature. The pellets were subsequently resuspended in the starting volume of lysis buffer. A polymerization reaction performed at 4°C was used as a negative control. The supernatants and pellets were analyzed by SDS-PAGE and immunoblotting.

## Quantification and statistical analyses

We define biological replicates as experiments that were performed on more than two separate days (usually one week apart) using cell lines at different passages. All data used for statistical analyses indicated in the legends to figures were acquired from at least three biological replicates. All graphs representing data were from at least three independent experiments. As indicated in the legends, statistical comparisons were made using either one-way analysis of variance (ANOVA) with Dunnett's multiple comparisons test, two-way ANOVA with Tukey's multiple comparisons test, or

Student's *t* test, as indicated by the Prism seven software. Numerical *p*-values are indicated in each graph; n.s. stands for not-significant differences.

## Acknowledgements

We thank Mikihiko Naito, David Rubinsztein, Didier Trono, Edward Yeh and Feng Zhang for kind gifts of reagents, Steven Coon and other members of the Molecular Genomics Core of NICHD for next-generation sequencing, Xiaolin Zhu for excellent technical assistance, and other members of the Bonifacino lab for helpful discussions.

## Additional information

### Funding

| Funder | Grant reference number | Author |
|---|---|---|
| NICHD | ZIA HD001607 | Juan S Bonifacino |

The funders had no role in study design, data collection and interpretation, or the decision to submit the work for publication.

### Author contributions

Rui Jia, Conceptualization, Resources, Data curation, Software, Formal analysis, Validation, Investigation, Visualization, Methodology, Writing—original draft; Juan S Bonifacino, Conceptualization, Resources, Supervision, Funding acquisition, Project administration, Writing—review and editing

### Author ORCIDs

Rui Jia (iD) https://orcid.org/0000-0002-1797-4069
Juan S Bonifacino (iD) https://orcid.org/0000-0002-5673-6370

### Decision letter and Author response

Decision letter https://doi.org/10.7554/eLife.50034.030
Author response https://doi.org/10.7554/eLife.50034.031

## Additional files

### Supplementary files

• Supplementary file 1. Summary of next generation sequencing reads in the primary genome-wide screen.
DOI: https://doi.org/10.7554/eLife.50034.020

• Supplementary file 2. MAGeCK of results from the primary genome-wide screen.
DOI: https://doi.org/10.7554/eLife.50034.021

• Supplementary file 3. List of sgRNAs used in the secondary CRISPR library.
DOI: https://doi.org/10.7554/eLife.50034.022

• Supplementary file 4. Summary of next generation sequencing reads in the secondary screen.
DOI: https://doi.org/10.7554/eLife.50034.023

• Supplementary file 5. MAGeCK of results from the secondary screen.
DOI: https://doi.org/10.7554/eLife.50034.024

• Supplementary file 6. List of sgRNAs used in the ubiquitination CRISPR library.
DOI: https://doi.org/10.7554/eLife.50034.025

• Supplementary file 7. Summary of next generation sequencing reads in the ubiquitination screen.
DOI: https://doi.org/10.7554/eLife.50034.026

• Supplementary file 8. MAGeCK of results from the ubiquitination screen.
DOI: https://doi.org/10.7554/eLife.50034.027

• Transparent reporting form DOI: https://doi.org/10.7554/eLife.50034.028

## Data availability

All data generated or analysed during this study are included in the manuscript and supporting files.

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
