## [Decision Letter]

**Acceptance summary:**

This work provides interesting evidence that the ubiquitin conjugation machinery proteins UBA6 and BIRC6 facilitate monoubiquitination of LC3 family proteins (not GABARAP proteins), followed by their proteasomal degradation. Cells depleted of UBA6 or BIRC6 have higher levels of non-lipidated LC3 (LC3-I) than control cells and increased autophagic flux when subjected to starvation or proteotoxic stress. UBA6 and BIRC6 were identified in two independent CRISPR/Cas9 knockout screens, targeting the whole-genome or genes involved in ubiquitin conjugation (E1, E2, E3) or deubiquitination, respectively, using H4 human neuroglioma cells expressing endogenous LC3B tagged with a tandem GFP-mCherry tag. The screens were based on FACS sorting of cells with increased GFP:mCherry ratio under basal conditions, thus selecting for positive regulators of autophagy. The identification of UBA6 and BIRC6 as negative regulators of autophagy by mediating degradation of LC3B is potentially therapeutically relevant as inhibition of UBA6/BIRC6 could be used to enhance autophagy.

**Decision letter after peer review:**

Thank you for submitting your article "Negative regulation of autophagy by UBA6-BIRC6-mediated ubiquitination of LC3" for consideration by *eLife*. Your article has been reviewed by three peer reviewers, including Ivan Dikic as the Reviewing Editor and Reviewer #1, and the evaluation has been overseen by Cynthia Wolberger as the Senior Editor. The following individual involved in review of your submission has agreed to reveal their identity: Christian Münch (Reviewer #2).

The reviewers have discussed the reviews with one another and the Reviewing Editor has drafted this decision to help you prepare a revised submission.

Essential revisions:

1) Presented data indicate a role for UBA6/BIRC6 in ubiquitination of LC3 is strong, but it is questioned whether accumulation of LC3-I in UBA6/BIRC6 KO is (only) due to a defect in proteasomal degradation of LC3. Other possibilities would be: 1) accumulation of LC3 in the nucleus (as LC3 K51 acetylation regulates this), 2) accumulation of LC3 in protein aggregates, 3) increased LC3 mRNA levels, 4) increased binding to GABARAP proteins or even microtubules. These possibilities should be ruled out.

2) The clearance of protein aggregates (ALIS and α-synuclein) is increased in BIRC6 KO cells. To support this finding the authors need to show that this is autophagy dependent and not due to effects on aggregate formation.

3) The authors identified the LC3B ubiquitylation site at K51, which is a highly conserved lysine residue in all six LC3 family members. How do the authors explain the selectivity for LC3 ubiquitylation and not for GABARAPs? Is this due their different roles in autophagy or do BIRC6/UBA6 only bind to LC3s? They should check why LC3s are ubiquitinated and not the GABARAPs, as the K51 residue is conserved. It is interesting that ubiquitination of LC3 might be a way of regulating its interaction with specific LIR-containing proteins and the authors need to comment or check binding of K51R to GABARAP-specific LIR proteins.

---

## [Author Response]

Essential revisions:1) Presented data indicate a role for UBA6/BIRC6 in ubiquitination of LC3 is strong, but it is questioned whether accumulation of LC3-I in UBA6/BIRC6 KO is (only) due to a defect in proteasomal degradation of LC3. Other possibilities would be: 1) accumulation of LC3 in the nucleus (as LC3 K51 acetylation regulates this), 2) accumulation of LC3 in protein aggregates, 3) increased LC3 mRNA levels, 4) increased binding to GABARAP proteins or even microtubules. These possibilities should be ruled out.

The extraction of LC3B for analysis by SDS-PAGE and immunoblotting was done under denaturing conditions using lithium dodecyl sulfate (LDS) and heating at 95°C, so it is unlikely that the accumulation of LC3B-I in UBA6- and BIRC6-KO cells could be due to anything other than changes in protein levels. Moreover, the increased levels of LC3B-I upon incubation with MG132 support the conclusion that LC3B-I degradation is proteasomal. Nevertheless, we performed additional experiments to test the additional possibilities suggested by the reviewers.

1) A nuclear fractionation assay showed that ~50% of LC3B-I was in the nuclear fraction and ~50% was in the cytoplasmic fraction, as previously reported (Huang et al., 2015). KO of BIRC6 did not alter these percentages, although the overall levels of LC3B-I were higher than in WT cells in both fractions, as expected from the decreased degradation of LC3B-I in the KO cells. These results are shown in a new Figure 5—figure supplement 1C.

2) Fractionation into Triton-soluble and -insoluble fractions, similar to those performed in Figure 8D for α-synuclein in rat hippocampal neurons, showed that very little LC3B-I was present in aggregates in WT H4 cells. KO of BIRC6 increased this small amount proportionally to the increase in total LC3B-I levels. These results are shown in a new Figure 5—figure supplement 1B.

3) Real-time quantitative PCR (RT-qPCR) demonstrated that KO of UBA6 or BIRC6 did not significantly alter LC3B mRNA levels relative to WT cells, as shown in the new Figure 2—figure supplement 3D.

4) A microtubule co-sedimentation assay showed no association of LC3B with microtubules in WT or BIRC6-KO cells, as shown in the new Figure 5—figure supplement 1D.

All of these experiments support our original conclusion that increased levels of LC3B-I in UBA6- and BIRC6-KO cells detected in SDS-PAGE and immunoblot analyses reflect increases in total protein levels due to decreased proteasomal degradation.

2) The clearance of protein aggregates (ALIS and α-synuclein) is increased in BIRC6 KO cells. To support this finding the authors need to show that this is autophagy dependent and not due to effects on aggregate formation.

To demonstrate that increased clearance of ALIS and α-synuclein aggregates in BIRC6-KO cells is dependent on autophagy, we examined the effect of ATG7 KD on the clearance of those aggregates. As shown in the new Figure 7—figure supplement 2A-C, ATG7 KD restored accumulation of aggregates in BIRC-6 KO H4 cells.

In the new Figure 8—figure supplement 2A-C, we also show that treatment with bafilomycin A_1_ increases α-synuclein aggregates in Birc6-KD rat hippocampal neurons.

Together with the results shown in Figure 7—figure supplement 1F and 1G), these new results demonstrate that clearance of ALIS and α-synuclein aggregates is dependent on autophagy.

3) The authors identified the LC3B ubiquitylation site at K51, which is a highly conserved lysine residue in all six LC3 family members. How do the authors explain the selectivity for LC3 ubiquitylation and not for GABARAPs? Is this due their different roles in autophagy or do BIRC6/UBA6 only bind to LC3s? They should check why LC3s are ubiquitinated and not the GABARAPs, as the K51 residue is conserved. It is interesting that ubiquitination of LC3 might be a way of regulating its interaction with specific LIR-containing proteins and the authors need to comment or check binding of K51R to GABARAP-specific LIR proteins.

To address this comment, we performed co-precipitation experiments of FLAG-BIRC6 with six GFP-tagged LC3/GABARAP family proteins. As shown in the new Figure 6—figure supplement 1C, BIRC6 interacts more strongly with LC3 than with GABARAP family proteins, explaining the selectivity of BIRC6 for LC3 ubiquitination. Although the K51 residue shared by LC3 and GABARAP family proteins contributes to some extent to interaction of LC3B with BIRC6 (new Figure 6—figure supplement 1D), other residues that differ between LC3 and GABARAP family proteins may contribute to the differential interactions. A structural explanation for these differences is beyond our possibilities at the present time.